# Water transport among the world ocean basins within the water cycle

David García-García[1], Isabel Vigo[1], Mario Trottini[2]

[1]Applied Mathematics Department, University of Alicante, San Vicente del Raspeig, 03690, Spain
[2]Mathematics Department, University of Alicante, San Vicente del Raspeig, 03690, Spain

*Correspondence to*: David García-García (d.garcia@ua.es)

**Abstract.** Global water cycle involves water-mass transport on land, atmosphere, ocean, and among them. Quantification of such transport, and especially its time evolution, is essential to identify footprints of the climate change and helps to constrain and improve climatic models. In the ocean, net water-mass transport among the ocean basins is a key, but poorly estimated parameter presently. We propose a new methodology that incorporates the time-variable gravity observations from the GRACE

satellite (2003-2016) to estimate the change of water content, and that overcomes some fundamental limitations of existing approaches. We show that the Pacific and Arctic Oceans receive an average of 1916 (95% confidence interval [1812, 2021]) Gt/month (~0.72 ± 0.02 Sv) of excess freshwater from the atmosphere and the continents that gets discharged into the Atlantic and Indian Oceans, where net evaporation minus precipitation returns the water to complete the cycle. This is in contrast to previous GRACE-based studies, where the notion of a seesaw mass exchange between the Pacific and

Atlantic/Indian Oceans has been reported. Seasonal climatology as well as the interannual variability of water-mass transport are also reported.

## 1 Introduction

The water-mass transport (henceforth WT for brevity) in the oceans is a deciding factor of the world climate system. Quantification of such transport, and especially its time evolution, is essential to better understand the climate change. Atlantic

Ocean presents notably a deficit of freshwater flux, in contrast to the Pacific Ocean. This produces a salinity asymmetry that explains why deep waters are formed in the North Atlantic and not in the North Pacific (Warren, 1983; Broecker et al., 1985; Rahmstorf, 1996; Emile-Geay et al., 2003; Czaja, 2009). Upper layers of North Atlantic flow northward, while deep waters flow southward, forming the Atlantic Meridional Overturning Circulation (AMOC), which distributes heat within the Earth system and influences temperature and precipitation patterns worldwide (Vellinga and Wood, 2002). While small changes in

hydrological cycle may have caused changes in AMOC during the last glaciation that led to abrupt climate changes (Clark et al., 2002), different models project a weakening of the AMOC in the 21[st] century that would lead to profound climatic and ecological changes (Collins et al., 2019). The Antarctic Circumpolar Current (ACC) receives deep water injected by AMOC with excess salinity, which in turn gets transported into the Indian and Pacific Oceans (Warren, 1981). The Indian Ocean returns saltier water, but Pacific and Arctic Oceans return less-salty waters, producing a salinity imbalance in

the Atlantic. To restore the balance, freshwater must be transported outside the Atlantic at the rate of 0.13-0.32 Sv through the atmosphere (Zaucker et al., 1994). This WT produces an excess of freshwater in other ocean regions, as in the Pacific and Arctic Oceans, that must discharge out through the ocean.

    Meanwhile, conventional observations on the lateral WT of world ocean climatology have been sparse. In fact, measuring such WT in an open ocean region proves difficult as it amounts only to a few tenths Sv, several orders of magnitude

smaller than the total ocean water inflow/outflows in such regions. For example, the Pacific is believed to receive regularly an inflow of $157 \pm 10$ Sv to south of Australia (Ganachaud and Wunsch, 2000), against three outflows: 0.7-1.1 Sv through the Bering Strait (Woodgate et al., 2012), $16 \pm 5$ Sv through the Indonesian Strait (Ganachaud and Wunsch, 2000), and 140-175 Sv through the Drake Passage (Ganachaud and Wunsch, 2000; Donohue et al., 2016; Colin de Verdière and Ollitrault, 2016; Vigo et al., 2018).


    In this work we propose a new methodology devised to estimate the net WT through the boundaries of a given oceanic region. A defining feature of the proposed approach is the use of the time-variable gravity data from the GRACE (Gravity Recovery and Climate Experiment) satellite mission to estimate the change of water content. We apply the methodology, in conjunction with conventional meteorological data of general hydrologic budget schemes, to estimate the time evolution over

the period 2003-2016 of the net WT and exchanges among the four major ocean basins – namely Pacific, Atlantic, Indian, and Arctic. We analyse and report our results of the seasonal climatology as well as the interannual variability of WT. Such information, not available previously, is of valuable importance. For example, in closed regions, net WT through the boundaries on the surface must be counteracted by moisture fluxes through the same boundaries in the atmosphere to match GRACE measurements. Such approach has been successfully applied to study the hydrological cycle of South America (Liu

et al., 2006). At ocean basin scale, knowledge about net WT not only would help elucidate the role of the oceans within the water cycle, but it will also impose restrictions on moisture advection in the atmosphere that would help to improve atmospheric models. On the other hand, ocean models usually deal with inflows and outflows of a given ocean region (Warren, 1983; Rahmstorf, 1996; Emile-Geay et al., 2003; de Vries and Weber, 2005; Dijkstra, 2007). Net WT estimates for such ocean region would be useful to impose constraints to the relationship between its inflows and outflows, which would improve the

reliability of the models. Better models will improve our knowledge of the Earth's WT dynamics and its evolution in the future, which is critical in the present scenario of climate change.

## 2. Methodology and Data

### 2.1 Methodology

    The general hydrologic budget equation states that, at any continental location and any moment in time, the change

of water content $dW$ equals the precipitation $P$ minus evapotranspiration $E$ (as vertical transport) minus the net runoff $R$ (as horizontal transport):

$$dW = P - E - R \qquad \text{for land.} \qquad (1)$$

Under the conservation of water mass, the global net $P-E$ over ocean is negative [e.g., Hartmann, 1994]. That amount of water gets transported to land through atmosphere and returns to the ocean as $R$ completing the water cycle. The general $R$ for a river basin connected to the ocean consists of river runoff, land ice melting, and submarine groundwater discharge to ocean. The $R$ component will be estimated as a residual in Equation 1.

For an ocean region, $R$ represents the inflow from adjacent land regions plus an extra additive term, call it $N$, accounting for water exchange between neighbouring ocean regions through boundaries, as (positive) inflow or (negative) outflow:

$$dW = P - E + R + N \qquad \text{for ocean.} \qquad (2)$$


The ocean water flux $N$ is the target quantity that we shall solve for as a residual in Equation 2, which up till now has been infeasible to estimate directly [Rodell et al., 2015]. Note that $N$ represents the integrated WT over the total-column depth of ocean, including deep-water flows. This is a strength of the GRACE observation for the oceans, compared to in-situ or other remote-sensing measurements typically targeting only the surface layer.


Our targeted four ocean basins are largely separated geographically with designated continental boundaries and restricted water throughways. The land is divided into their associated drainages according to the global continental runoff pathways scheme of Oki and Sud (1998). There are no direct water exchanges in the form of $R$ among land drainages (see Figure 1). The WT component $R$ is estimated through Equation (1) over each continental region, then input to Equation (2) to
estimate $N$ in the associated ocean basins.

**2.2 Precipitation and Evaporation data**
The $P$ and $E$ data we use are adopted from the ERA5 reanalysis [Hersbach et al., 2018], which assimilates observations into general-circulation modelling provided by the European Centre for Medium-Range Weather Forecasts (ECMWF). They
are given at 0.25º latitude/longitude regular grids and monthly (and hourly) intervals for global coverage of both continents and oceans. In order to match the spatial resolution of the above-mentioned continental runoff pathways data, we homogenise the grid to 1ºx1º by averaging the corresponding 0.25º grid points.

**2.3 Time-variable GRACE data**

The critical knowledge needed in Equations (1) and (2), now obtainable from GRACE monthly data, is $dW$ (Tapley et al., 2004, 2019), the month-to-month difference of the stored water. Note that the GRACE mass variability pertains to WT directly, as opposed to, for example, altimetric sea level measurements that also contain non-WT, steric effects. We use the GRACE "mascon" (mass concentration) solutions that have already been converted into surficial mass from the original time-variable gravity observations (in our case the GRACE RL06 mascon dataset provided by the Center of Space Research (CSR) of University of Texas; see Save et al., 2016, Save, 2019). The non-surficial gravity change due to the glacial isostatic adjustment (GIA) has been removed to the extent of the ICE6G-D model (Peltier et al., 2018). Any other non-surficial effect such as long-term tectonics would be incorrectly interpreted as water mass fluxes (Chao, 2016) but they may only have importance in the determination of secular trends; so are the non-climatic sources such as the rare, local earthquake events. As the $C_{20}$ Stokes coefficient is not well determined from GRACE mission, it is replaced with a more accurate solution from Satellite Laser Ranging (SLR) (Cheng and Ries, 2017). GRACE is not sensitive to the geocenter variations, and its degree-1 Stokes coefficients are set to zero. We had tried adding to GRACE data an estimate of geocenter variations due to modelled water-mass variability (Swenson et al., 2008), and our reported results would change less than 1%. On the other hand, the atmospheric, and some oceanic, effects on gravity change had beforehand been removed from the processing of the GRACE data by CSR, for de-aliasing purposes, according to the operational numerical weather prediction (NWP) model from ECMWF and to an unconstrained simulation with the global ocean general circulation model MPI-OM -Max-Planck-Institute Global Ocean/Sea-Ice Model- (Dobslaw et al. 2017). To recover the "true" ocean mass variability, we restore the removed signal on the oceans adding back the GAD product, which is set to zero on the continents. Data are provided on a 0.25º regular grid; we reduce it to 1º regular grids, still finer than the spatial resolution of GRACE (~300 km), to match the spatial resolution of the continental drainage basin data as above.

GRACE's degree-0 Stokes coefficients $\Delta C_{00}$ is set identically to zero on the recognition that Earth's total mass (including the atmosphere) is constant. Then, any increase (decrease) of the water-mass budget of the atmosphere will be counteracted by a decrease (increase) of the same amount of water-mass in the surface. However, after the atmospheric and dynamic oceanic mass changes are corrected in GRACE data, the GRACE $\Delta C_{00}$ are still set to zero even though they should match the opposite of the removed signals. To restore the lost degree-0 signal, the GAD product (which is set to zero on the continents) must be added back to GRACE with averaged ocean signal set to zero, and then, the $\Delta C_{00}$ from an atmospheric model must be subtracted from GRACE data to force the Earth's total mass to be constant. Doing so, the GRACE data will account for the global exchange of water-mass between the Earth surface and atmosphere. Such correction has recently proved to improve the agreement between the GRACE global ocean mass change and non-steric sea level variation estimates from altimetry and ARGO data (Chen et al., 2019). Looking for consistency between the GRACE and ERA5 datasets, we use $\Delta C_{00}$ from $P-E$ to restore degree-0 signal in $dW$. This $\Delta C_{00}$ accounts for uniform mass variations in the global surface equivalent to a global averaged signal for $P-E$, at 188 Gt/month (95% confidence interval $CI_{95}$=[136, 243], see below). As global $-(P-E)$

represents the variability of global total-column water (TCW), it should match the time derivative of the global TCW. However, the average rate of change of the global TWC in ERA5 is 1.5 Gt/month (CI$_{95}$=[−9.2, 12.7]), although in the range of previously reported values of [−0.9, 4.3] Gt/month [Nilsson and Elgered, 2008] departs far from the global –$(P−E)$ value. This reveals some internal inconsistency within the ERA5 dataset. However, while artificially increasing the $dW$ estimate, the excessive value of $P−E$ does not affect the WT components $R$ and $N$ estimated from Equations (1) and (2), since the degree-0 signal vanishes due to the residual estimate between $dW$ and $P−E$. In fact, adding $\Delta C_{00}$ from $P−E$ to GRACE is numerically equivalent to setting $P−E$ $\Delta C_{00}$ to zero as far as Equations (1) and (2) are concerned.

## 2.4 Confidence intervals

The reported 95% confidence intervals and the correlation coefficients are evaluated using the stationary bootstrap scheme of Politis and Romano (1994) (with optimal block length selected according to Patton et al., 2009), and the percentile method. The intuition underlying the bootstrap is simple. Suppose that the observed time series $x_1, ..., x_n$ is a realization of the random vector $(X_1,..., X_n)$ with joint distribution P$_n$ and which is assumed to be part of a stationary stochastic process. Given $X_n$, we first build and estimate $\hat{P}_n$ of $P_n$. Then $B$ random vectors $(X_1^*, ..., X_n^*)$ are generated from $\hat{P}_n$. If $\hat{P}_n$ is a good approximation of P$_n$, then the relation between $(X_1^*, ..., X_n^*)$ and $\hat{P}_n$ should well reproduce the relation between $(X_1,..., X_n)$ and $P_n$ (for an introduction of bootstrap methods for time series see Kreiss and Lahiri (2012) and the references therein). Here, the number of bootstrap replications was set to $B$=2000. In general, half length of the confidence interval can be very well approximated by twice the standard deviation of the sample mean estimated from the bootstrap replications. Prior to applying the bootstrap to a time series, least-squares estimated linear/quadratic trend and sinusoid with the most relevant frequencies are removed from it to meet the stationarity conditions of the method. In particular, each series has been decomposed into trend, seasonal and residual components. The bootstrap is applied to the residual component producing bootstrap samples of the residuals. For the evaluation of confidence intervals for the different components of WT, the trend and seasonal terms are added back (to the bootstrap sample of the residuals) producing bootstrapped time series of the component of interest. These samples are then used for further analysis. As an illustration, for the WT $N$ component we proceed as follows: (i) a model with linear, annual, and semiannual signals is fitted to the data. The fitted linear trend and annual and semiannual signals are subtracted from the original time series; (ii) the stationary bootstrap is then applied to the residuals producing 2000 bootstrap samples of the residuals; (iii) The estimated trend and seasonal components are added back to each bootstrap sample of the residuals obtaining an ensemble of 2000 bootstrapped time series for the $N$ component; (iv) these 2000 bootstrapped time series are used to obtain 95% confidence intervals for the mean fluxes (average of $N$ over the 14 year period of study) and for the amplitude and phase of the annual component using the percentile method. For the mean fluxes, the average of $N$ for each of the 2000 bootstrapped time series was first evaluated and then the 0.025 and 0.975 percentiles of these 2000 averages were reported as 95% confidence interval. For the study of the climatology, a linear trend model with annual and semiannual components was fitted to the 2000 bootstrapped time series producing corresponding estimates of the annual amplitude and

phase. The 0.025 and 0.975 percentiles of these estimates were reported as 95% confidence intervals. In order to study the robustness of the results with respect to the model choice, the analysis is rerun using 11 alternative models obtained considering different forms for the trend component (quadratic or constant) and including higher frequencies in the harmonic regression (up to 5). The results are robust. The relative difference with respect to the reported values is smaller than 1.2% for point estimates and smaller than 3.3% for the extremes of the 95% confidence intervals.

As an independent check of the bootstrap, confidence intervals for the mean value of $N$ have been also evaluated by propagating the error estimate in GRACE data (using the JPL GRACE mascon solution for which error estimates are available). The resulting intervals were consistent with those of the bootstrap method. In particular (see Section 4 for details), we show that in all cases the bootstrap intervals contain the intervals obtained from error propagation. In this respect, the $CI_{95}$ from bootstrap analysis can be considered a conservative estimate. This should be expected, since the residual component underlying the bootstrap approach includes measurement errors and other type of errors (related, for example, with the estimate of the trend and seasonal terms). As a result, the uncertainties in the transports estimated by the bootstrap should be larger than the corresponding uncertainties estimated by error propagation.

Note that for the study of correlation the bootstrap was applied to the bivariate time series of the residuals of the two variables of interest producing an ensemble of 2000 bivariate time series of residuals. For each bivariate time series of residuals the correlation between the two components of the series was first evaluated. The average and the 0.025 and 0.975 percentiles of these 2000 estimates were reported as point estimate and confidence limits for the correlation between the two variables of interest (correlation between residual components is used to avoid spurious correlation).

## 3. Results

The various WT components of the Pacific and its associated land drainage regions are shown in Figure 2 in units of Gt/month (1 Sv ≈ 2600 Gt/month; 1 Gt = $10^{12}$ kg, the weight of 1 km$^3$ of freshwater). The same analysis is applied to the rest of the ocean basins, i.e. the AIA oceans individually and collectively, with its associated land drainages (see Figure 1).

### 3.1 Mean fluxes

Averaged over the studied 14 years, the Pacific Ocean loses water through the atmospheric $P-E$ at the average rate of 142 Gt/month ($CI_{95}$=[48, 243]), which is greatly over-compensated by inflow $R$ from land of 1403 Gt/month ($CI_{95}$=[1370, 1436]). From this surplus, a minor (if any) amount of 67 Gt/month ($CI_{95}$=[25, 108]) stays (and accumulates) in the Pacific, while 1194 Gt/month ($CI_{95}$=[1096,1291]) is transported horizontally to the "non-Pacific" Atlantic/Indian/Artic (AIA) oceans, which will be called the "Pacific outflow" hereafter.

In the AIA Oceans, the situation is found to be markedly distinct, given the fact that the AIA oceans together have surface area comparable to the Pacific ($177 \times 10^6$ m$^2$). The AIA oceans collectively lose 3484 Gt/month (CI$_{95}$=[3406, 3560]) through the atmospheric $P-E$, that is ~25 times more than does the Pacific. This water deficit is only ~68% compensated by land $R$ inflow of 2378 Gt/month (CI$_{95}$=[2337, 2419]). With the nominal minor amount of water accumulation at 87 Gt/month (CI$_{95}$=[44, 130]), the AIA oceans thus presents an average inflow of 1194 Gt/month (CI$_{95}$=[1102, 1284]) from the Pacific, which will be called the "AIA inflow".

As expected from the overall conservation of water mass inherent in our methodology, the estimated Pacific outflow and AIA inflow match (Figure 3). It is worth mentioning that a difference of 188 Gt/month would exist between the two mean flux values if the degree-0 correction were not applied.

Corresponding analyses have been performed for the Atlantic, Indian, and Arctic Oceans separately. The time evolution of the WT components in Eqs. 1 and 2 are shown in Figure 4, and a diagram of the water-mass fluxes is shown in Figure 5. On average, the Atlantic Ocean receives 926 Gt/month (CI$_{95}$=[876, 980]; or 0.36 Sv) of salty water, and loses to the atmosphere 879 Gt/month (CI$_{95}$=[828, 930]) via $P-E+R$. The latter is equivalent to a freshwater deficit of 0.34 Sv, which increases the near-surface salt concentration and enables water to sink in North Atlantic producing deep water. These values are close to the 0.13-0.32 Sv estimated from ocean models, as needed to keep salinity balance in the Atlantic Ocean (Zaucker et al., 1994). Similarly, the Indian Ocean loses 957 Gt/month (CI$_{95}$=[894, 1022]) of freshwater that is restored by 991 Gt/month (CI$_{95}$=[907, 1073]) of salty water. The freshwater lost via $P-E+R$ by the Atlantic and Indian Oceans goes to the Pacific (1261 Gt/month, CI$_{95}$=[1171, 1347]) and Arctic (730 Gt/month, CI$_{95}$=[712, 747]) Oceans, which return 1194 (CI$_{95}$=[1096, 1291]) and 723 (CI$_{95}$=[708, 739]) Gt/month of salty water through the ocean, respectively. Then, the Pacific presents a surplus of freshwater that reduces near-surface salt concentration, which prevents the formation of deep water. Together, the Pacific and Arctic Oceans supply 1917 Gt/month (CI$_{95}$=[1812, 2021]) of water to the Atlantic and Indian Oceans, where it is reincorporated into the water cycle via net $E-P$. As in previous studies (see Craig et al., 2017 for a synthesis), the freshwater lost in the Indian Ocean is similar to that in the Atlantic Ocean. In these studies, $P-E+R$ is close to zero in the Pacific Ocean, producing a difference of 0.4 Sv between Atlantic and Pacific Oceans. In our study, $P-E+R$ is 1261 Gt/month in the Pacific Ocean and the difference with the Atlantic increases to ~0.8 Sv. Some of these differences would be expected as far as the ocean basins are not defined in exactly the same way. On the other hand, the global $R$ is 3781 Gt/month (or $3781 \times 12 = 45368$ km$^3$/year), close to the 41867 km$^3$/year reported by the Global Runoff Data Centre (GRDC, 2014). At basin scale, $R$ is 16834 km$^3$/year in the Pacific, greater than the 11826 km$^3$/year reported by GRDC. In the Atlantic, Indian, and Arctic, $R$ is 18228, 4479, and 5827 km$^3$/year, respectively, which is closer to the GRDC values: 20772, 5238, and 4080 km$^3$/year. Finally, according to the diagram in Figure 5, the water content in the atmosphere decreases 178 Gt/month (and it is gained by Earth's surface), but this amount is not realistic as discussed in Section 2 since it should increase a few Gt/month (Nilsson and Elgered, 2008). This value differs from the 188 Gt/month mentioned in Section 2 because the endorheic regions are not accounted here.

## 3.2 Annual climatology

The WT climatology of the $N$ component is estimated in two ways: (1) averaging the 14 $N$ values for each months of the year (Figure 6a); and (2) fitting a linear trend plus annual and semiannual components model as described in Section 2. Annual amplitudes and phases  are plotted in Figure 6b and reported, with corresponding 95% quantile-based confidence intervals, in Table 1.

The Pacific and Arctic Oceans show an overall outflow throughout the year, unlike the Atlantic and Indian Oceans, which show an inflow for every month. The Pacific outflow shows a prominent seasonal undulation peaked around August 3 and a peak-to-peak WT variation of ~2000 Gt/month from boreal summer to November, when a near-zero minimum occurs. The Arctic Ocean show half of the Pacific variability and a less pronounced seasonal undulation. A minimum outflow of ~320 Gt/month is reached in March and April, and a maximum ~1300 Gt/month in July. Together, the Pacific and Arctic Oceans

send ~3000 Gt/month of seawater to the Atlantic and Indian Oceans during boreal summer, and a minimum amount five times lower, around 600 Gt/month, in November. The annual maximum is reached on August 8th. The Atlantic/Arctic inflow mirrors this behaviour. Separately, the Atlantic and Indian inflows show a similar peak-to-peak variation of ~2000 Gt/month, reaching the maxima in August and May, respectively. The Indian maximum seems to be related to a local maximum of the Pacific outflow. The annual maxima of net WT of the four basins are reached between August 3rd and September 9th , although the

annual signals of the Pacific and Indian Oceans almost triple those from Arctic and Atlantic Oceans (Table 1 and Figure 6b).

## 3.3 Interannual variability

       Interannually, the Pacific outflow shows remarkable variability, mainly produced by $P$ on the continents, which is inherited by $R$, and $P-E$ in the oceans (Figure 2). For example, the Pacific outflow shows a maximum around 1372 Gt/month

in 2009 that matches with a $P-E$ maximum in the Pacific, $P-E$ minimum in the AIA oceans, and $P$ minima in the continental basins draining to both Pacific and AIA oceans. The opposite behaviour, that is a minimum around 939 Gt/month is observed in 2010. The difference, 433 Gt/month, is comparable to the discharge of Amazon (Lorenz et al., 2014). In the tropical Pacific, the El Niño/Southern Oscillation (ENSO) is the strong recurring climate pattern involving changes in the temperature of seawater and air pressure in the tropical Pacific Ocean. The ENSO had a mild El Niño phase in 2009 followed by a strong La

Niña phase in 2010, that may be related to the interannual variability of the Pacific outflow. To elucidate this, we conduct a correlation study of the interannual Pacific outflow with respect to the major climate oscillations in the Earth's atmosphere-ocean: ENSO, Atlantic Multi-decadal Oscillation (AMO), Antarctic Oscillation (AAO), and Arctic Oscillation (AO). The climatic oscillation is represented by monthly time series of its indices, which are non-dimensional functions of time derived

from relevant meteorological observations; their values indicate the polarity and strength of the oscillation at a given epoch.

The ENSO oscillations are measured here with the Southern Oscillation Index (SOI), which represents the sea level pressure differences between Tahiti and Darwin, Australia. The AMO is a coherent mode of natural variability based upon the average anomalies of sea surface temperatures, with AMO Index to reflect the non-secular multi-decadal sea surface temperature pattern variability in the North Atlantic basin. The AAO describes the intensity of westerly wind belt surrounding the Antarctic, quantified by the AAO Index, which is the leading principal component of the 700 hPa atmospheric geopotential height

anomalies poleward of 20°S. The AO is to be interpreted as the surface signature of modulations in the strength of the polar vortex aloft the Arctic, while the AO Index is constructed by projecting the 1000 hPa height anomalies poleward of 20°N. Figure 7a show all indices with amplitudes normalized to one standard deviation, as well as the de-trend, de-season, standard deviation normalized Pacific outflow. The correlation analysis between the Pacific outflow and the SOI shows no overall correlation (Pearson coefficient of 0.03) during the whole period, meaning that the influence of ENSO on the Pacific outflow

may be restricted to the strong phases of ENSO as in 2009 and 2010. A similar lack of correlation (lower than 0.1) is observed with respect to the AMO, AAO, and AO.

To explore this lack of correlation, we have estimated the correlation coefficient between each climatic index and each WT component (Figure 7b). All of them are lower than 0.3 except for 6 cases in 2 regions. In the Arctic, $P$ and $P-E$ in

the drainage basins of the Arctic show a correlation of ~0.5 with the AO. This correlation is natural since that is the area of influence of the AO. The other region is the Pacific, where, as expected, the SOI shows a correlation around 0.5 with $P$, $P-E$, and $R$ in the drainage basins, and around $-0.4$ with $P$ in the ocean. However, this individual correlation does not extend to the Pacific outflow. In order to understand why this is the case, it is convenient to express the $N$ component of the water transport as a function of $P-E$ and $dW$. According to Equations 1 and 2 we have:


$$N = -(P-E)_{ocean} - R + dW_{ocean} = \underbrace{-(P-E)_{ocean}}_{X_1} \underbrace{-(P-E)_{land}}_{X_2} + \underbrace{dW_{land}}_{X_3} + \underbrace{dW_{ocean}}_{X_4}. \qquad (3)$$

It can be shown that the correlation between $N$ and a given index can be express as follows


$$corr(N, SOI) = \sum_{i=1}^{4} corr(X_i, SOI) \cdot \frac{std(X_i)}{std(N)}, \qquad (4)$$

where corr denotes the correlation coefficient, and std stands for standard deviation. As shown in equation (4), the correlation between $N$ and a given index is a linear combination of the correlation between each component and the index. The coefficients of the linear combination, $std(X_i)/std(N)$, are proportional to the standard deviation of each component. The components of

equation (4) for the Pacific outflow and the SOI index are shown in Table 3. Despite the fact that some of the individual component exhibits significant correlation with SOI (in particular $P-E$ in land and ocean) when combined with the

corresponding coefficients their effects canceled out yielding a negligible correlation between water transport and SOI (below 0.03 in magnitude). Note that table 3 provides also some insights about the causes of the interannual variability of Pacific Ocean outflow. The largest standard deviation of $P-E$ and $dW$ in the ocean suggests that these two components might drive the interannual variability of the Pacific Ocean outflow. This is confirmed by a correlation analysis. The correlation between $N$ and $(P-E)_{ocean}$ is -0.70. The correlation between $N$ and $dW_{ocean}$ is 0.84. The correlation of $N$ with the corresponding land components is below 0.18. In all cases, prior to the evaluation of the correlation the corresponding time series have been de-trend and de-season.

Another possible reason for the lack of correlation resides in the definition of the studied regions, for which the presence of subregions with positive and negative influence of an index results in an overall negligible/attenuated influence of the index in the overall region. For example, a positive phase of the AMO is related to an increase of $P$ in western Europe (Sutton and Hodson, 2005), and the Sahel (Folland et al., 1986; Knight et al., 2006; Zhang and Delworth, 2006; Ting et al., 2009), but to a decrease of $P$ in the U.S. (Enfield et al., 2001; Sutton and Hodson, 2005), and northeast Brazil (Knight et al., 2006; Zhang and Delworth, 2006). All these regions are included in the Atlantic drainage basin, and then the influence of a positive phase of the AMO is attenuated.

## 4. Comparison with other datasets

In this section, we perform a comparisons using alternative datasets. In particular:

(1) CSR GRACE mascon solution is replaced by the JPL GRACE mascon solution provided by the Jet Propulsion Laboratory/NASA (Watkins et al., 2015; Wiese et al., 2019). Similarly to CSR data, JPL are corrected for GIA effects, $C_{20}$ Stoke coefficients are replaced by a solution from SLR, and data are reduced to 1° regular grids from 0.5° regular grids. Besides, we have applied the degree-0 Stoke coefficients correction. However, CSR and JPL mascon solutions are not directly comparable. The main reason is that an estimate of degree-1 coefficients has been added to JPL mascon solutions, and the GAD product has not been added back. The corrections applied by JPL are not supplied separately and we cannot do/undo any of the corrections to process JPL data as we did with CSR data. In particular, the GAD product is not available for JPL. In any case, the JPL solution is useful here since it provides an error estimate of the mascon solution that can be propagated to obtain confidence intervals of $N$, which are independent from those estimated with the bootstrap analysis. Table 2 shows the CI$_{95}$ of the mean values of the $N$ component for different ocean basin estimated from error propagation and bootstrap analysis. It is observed that in all cases the CI$_{95}$ from error propagation are included in those from bootstrap analysis, meaning that the latter are a conservative estimate of the error. JPL propagated error can be expected to be similar to that propagated from CSR error estimates (which are not available), and then we can assume that the reported CI$_{95}$ for $N$ calculated from CSR data are a conservative estimate. Besides, comparing Tables 1 and 2, it is observed that the mean values of $N$ are quite similar and that the CI$_{95}$ largely overlap. Regarding to the time variability,

the values of the *N* component from CSR and JPL mascon solutions show Pearson correlation coefficients greater than 0.85 (p-value $< 10^{-3}$), except for the Atlantic (0.70). Thus, despite the different processing of CSR and JPL data, the reported analysis for the *N* component is robust with respect to the choice of GRACE datasets.

(2) ERA5 *P* and *E* data are replaced by several datasets for comparison purposes. The objective is not to be exhaustive in the selection, but rather to show that the reported features of the *N* component are quite robust with respect to the choice of the *P* and *E* datasets. The data sets considered are:

(i) Continental *P* from GPCC (Schneider et al., 2011), GPCP (Adler et al., 2018), CMAP (Xie and Arkin, 1997), UDel (Willmott and Matsuura, 2001), and GLDAS/Noah (Rodell et al., 2004; Beaudoing and Rodell, 2016).

(ii) Ocean *P* from GPCP and CMAP.

(iii) Continental *E* from GLEAM (Miralles et al., 2011; Martens et al., 2017) and GLDAS/Noah.

(iv) Ocean *E* from OAFlux (Yu et al., 2008) and HOAPS/CM SAF (Schulz et al., 2009).

The Pacific outflow is estimated with the 162 possible combinations of *P* and *E*, including ERA5. The time period is 2003-2016, except for HOAPS/CM SAF and GPCP, which span from 2003 to 12/2014 and 10/2015, respectively. The degree-0 corrections in GRACE data is made for each combination. Note that only ERA5 includes *P* and *E* for both continents and oceans. All grids have been homogenized to 1° regular grids. The main concern here is the heterogeneity of the spatial coverage among datasets. To make the results comparable among datasets, the computations are restricted to the common grid points, which do not cover the entire Earth (Figure 8a). However, in spite of the fact that due to the partial coverage the principle of water mass conservation is not accomplished, the Pacific outflow obtained in the common grid points from ERA5 (black line in Figure 8b) is quite in agreement with the same signal obtained with global coverage (red line in Figure 3 which is also reported as red line in Figure 8b). The Pearson correlation coefficient between the two signals is 0.994 (p-values $< 10^{-3}$) with an average difference around 50 Gt/month. In general, the Pacific outflows estimated from all the *P* and *E* dataset combinations show qualitatively the same signal than the one reported in Figure 3. For each of the 162 estimates of the Pacific outflows corresponding to the possible *P* and *E* dataset combinations, we evaluated the average outflow (over the period of study), which is 968 Gt/month (STD: 489), and the correlation with the Pacific outflows in Figure 3, which is 0.82 (STD: 0.06; p-values $< 10^{-3}$).

These experiments show that the reported net WT are physically consistent among datasets, at least qualitatively.

**5. Discussion and Conclusions**

In this work we present a new methodology that combines GRACE data with the general hydrologic budget equation to estimate the horizontal water-mass convergence/divergence for any oceanic region. We have assumed that the gravity changes are produced by mass changes on the Earth surface, such as in the oceans, so that the mascon solution is physically

meaningful (Chao, 2016). Any mis-modelling of the ocean basin "container" volume change due to GIA and other non-surficial changes would masquerade as WT variations. However, they are not critical as far as our non-secular analysis is concerned.

We use the proposed methodology to estimate the net WT and exchanges among the Pacific, Atlantic, Indian, and Arctic Oceans, for the period of 2003 – 2016. Our main finding is that the Pacific and Arctic Oceans, while replenished with
precipitation and land runoff, are nearly continuously losing water to the Atlantic and Indian Oceans. In particular, the WT climatology is such that the Pacific Ocean loses water at a rate from near zero to up to the peak of 2000 Gt/month during the boreal summer, which coincides with the maximum of the global atmosphere water content. On top of the climatology, the interannual Pacific water loss varies significantly between ~950 to ~1450 Gt/month annual means during the studied period, but seemingly uncorrelated with ENSO.


The results presented here are consistent with the well-known salinity asymmetry between the Pacific and Atlantic Oceans (Reid, 1953; Warren, 1983; Broecker et al., 1985; Zaucker et al., 1994; Rahmstorf, 1996; Emile-Geay et al., 2003; Lagerloef et al., 2008; Czaja, 2009; Reul, 2014). However, they are in contrast to previous GRACE-based studies where a simple seesaw WT between the Pacific and the Atlantic/Indian oceans was reported (Chambers and Willis, 2009; Wouters et
al., 2014). In those studies, the $P-E+R$ term in Equation 2 in both Pacific and Atlantic/Indian Oceans was approximated by that from the global ocean mean. However, the mean freshwater flux in the Pacific (1261 Gt/month) quite mis-matches that in the Atlantic/Indian Oceans (−1837 Gt/month), meaning that the approximation was quite poor and hence the $N$ term was not properly estimated in these studies (see Appendix for further discussion).

Differences in freshwater fluxes between the Pacific and Atlantic Oceans produce salinity contrasts, and in turn contrasts on deep water formation. Nevertheless, there are other factors influencing these contrasts such as the narrower extent of the Atlantic (de Boer et al., 2008; Jones and Cessi, 2017), the meridional span of the African and American continents (Nilsson et al., 2013; Cessi and Jones, 2017), and the salty WT from the Indian Ocean to the Atlantic (Gordon, 1986; Marsh et al., 2007). AMOC is also influenced by WT through Bering Strait (Reason and Power, 1994; Goosse et al., 1997; Wadley
and Bigg, 2002), and by surface processes of temperature, precipitation and evaporation at low-latitudes of Pacific and Indian Oceans (Newsom and Thompson, 2018). The relative importance among the multiple drivers influencing the AMOC is an open problem (Ferreira, 2018). The net WT estimated here provides information for differences between oceanic inflows and outflows, which can be useful to elucidate on this problem.

Net WT in the open oceans can alternatively be estimated using global ocean models, which simulate observational data based on physical principles. However, these models are not necessarily sensitive to the WT specifically given the data types, and the geography and topography resolutions involved in the models. Knowledge about three-dimensional global ocean

circulation could also elucidate on the net WT. However, the small ratio between the net and the total WT hinders the estimation of the former from the latter.


We have applied our WT estimation scheme to the four major ocean basins. The methodology can of course be applied to any extensive ocean region of interest as long as it is much larger than the GRACE resolution. The findings reported here will be useful for a better understanding of the global climate system in terms of its climatology and spatio-temporal variations.

**Appendix: Apparent net mass exchange between Pacific and Atlantic/Indian oceans**

We shall show here that the net water mass exchange between the Pacific and Atlantic/Indian Oceans reported by Chambers and Willis (2009) was a mathematical artefact. Their Equation (2) approximated the freshwater flux, i.e. $P-E+R$, of the Pacific (Pcf) and Atlantic/Indian (AI) oceans by the global ocean (GO) mean. However, from Figures 2 and 4 we get very different $(P-E+R)_{Pcf}=1261$ and $(P-E+R)_{AI}=-1837$ Gt/month, meaning that the approximation in Chambers and Willis

(2009) and hence their resultant estimates of WT are rather poor. In addition, under their approximation an apparent net mass exchange will always arise, since

$$
\begin{aligned}
(P-E+R)_{GO} &= \sum_{x \in GO} \frac{(P-E+R)_x \cdot Area(x)}{Area(GO)} = \\
&= \sum_{x \in Pcf} \frac{(P-E+R)_x \cdot Area(x)}{Area(GO)} + \sum_{x \in AI} \frac{(P-E+R)_x \cdot Area(x)}{Area(GO)} = \\
&= \sum_{x \in Pcf} \frac{(P-E+R)_x \cdot Area(x)}{Area(Pcf)} \cdot \frac{Area(Pcf)}{Area(GO)} + \sum_{x \in Atl/Indian} \frac{(P-E+R)_x \cdot Area(x)}{Area(AI)} \cdot \frac{Area(AI)}{Area(GO)} \approx \\
&\approx (P-E+R)_{Pcf} \cdot \frac{1}{2} + (P-E+R)_{AI} \cdot \frac{1}{2},
\end{aligned}
$$

where $x$ are disjoint grid cells in the ocean basins, the areas of the Pcf, AI, and GO are around 177, 160, and 351 x $10^6$ km$^2$, and the ratios 177/351 and 160/351 have been approximated by 1/2. Then, multiplying by 2 and rearranging the equation we get,

$$
(P-E+R)_{Pcf} - (P-E+R)_{GO} \approx -[(P-E+R)_{AI} - (P-E+R)_{GO}].
$$


Thus, wherever the signal is in the Pacific and Atlantic/Indian oceans, the anomalies with respect to the global ocean mean will always mirror each other, showing an apparent net mass exchange between them, even if such exchange does not exist.

## Data availability

All datasets used in this study are publicly available.

## Author contributions

DGG designed the study, processed datasets, and wrote the first draft. MT and IV evaluated the bootstrap confidence intervals, and contributed to other aspects of the data analysis including the robustness with respect to the use of alternative datasets. IV also provided funding for the research. All the authors discussed and interpreted the results, contributed to the writing and revision of the manuscript.

## Competing interests

The authors declare that they have no conflict of interest.

## Acknowledgements

We would like to thank Ben Chao for sowing the seeds of the idea of this work, during a research stay of D. García-García in Goddard Space Flight Center/NASA in 2004, and for all the subsequent fruitful discussions and advice. We thank the organizations that provide the data used in this work, which are publicly available: ERA5 data provided by the Copernicus Climate Change Service Climate Data Store (CDS), https://cds.climate.copernicus.eu/cdsapp#!/home; GPCC, CMAP, and UDel precipitation data provided by the NOAA/OAR/ESRL PSD, Boulder, Colorado, USA, https://www.esrl.noaa.gov/psd/; GPCP Precipitation data by the Mesoscale Atmospheric Processes Laboratory, NASA Goddard Space Flight Center, https://precip.gsfc.nasa.gov/; OAFlux evaporation data provided by the WHOI OAFlux project (http://oaflux.whoi.edu) funded by the NOAA Climate Observations and Monitoring (COM) program; HOAPS Evaporation data provided by CM SAF/EUMETSAT, https://www.cmsaf.eu; GLDAS/Noah data by GSFC/NASA, https://disc.gsfc.nasa.gov/; GLEAM evaporation data available at https://www.gleam.eu/; GRACE time-variable gravity data provided by CSR, University of Texas, http://www2.csr.utexas.edu/grace, and by JPL/NASA, https://podaac.jpl.nasa.gov/dataset/TELLUS_GRAC-GRFO_MASCON_CRI_GRID_RL06_V2; Southern Oscillation Index (SOI) from Bureau of Meteorology de Australia; Atlantic Multidecadal Oscillation (AMO) from the NOAA Earth System Research Laboratory (ESRL); Arctic Oscillation (AO) and Antarctic Oscillation (AAO) from NOAA Climate Prediction Center (CPC). This research is funded by Spanish Ministry of Science, Innovation and Universities grant number RTI2018-093874-B-100.

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

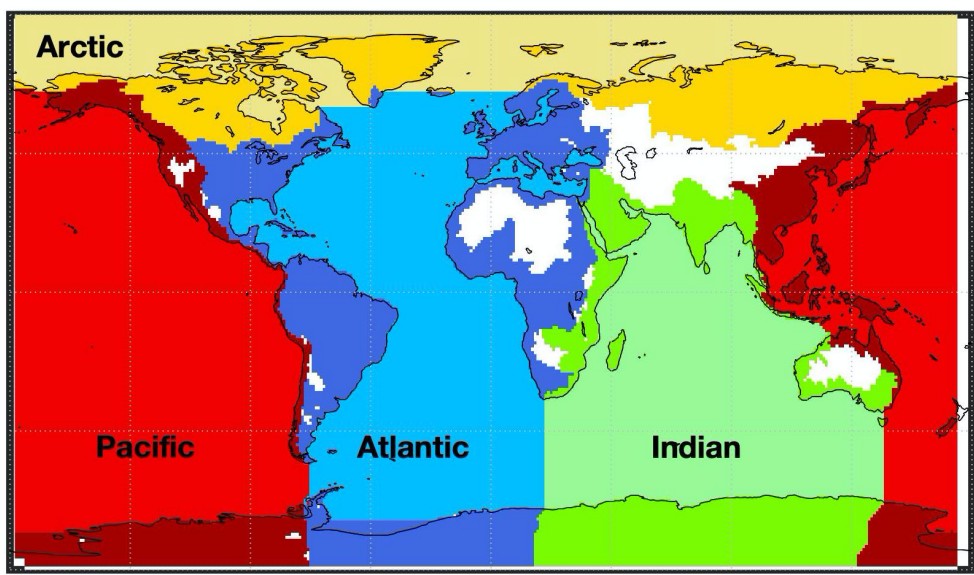

**Figure 1.** Pacific, Atlantic, Indian, and Arctic Ocean basins and their associated continental drainage basins according to the global continental runoff pathways scheme of Oki and Sud (1998). Within each basin, darker colour represents the continental basin, lighter colour the ocean basin. White regions represent endorheic basins.

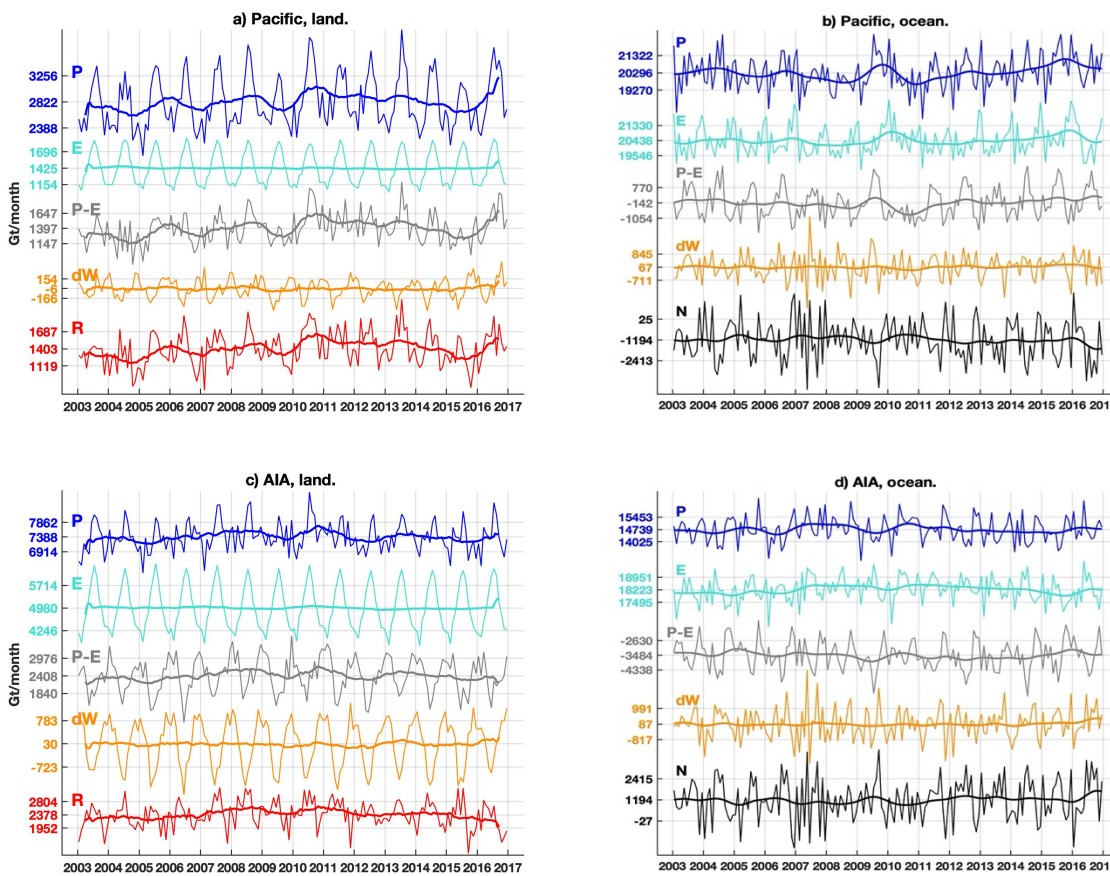

**Figure 2. WT of Equations (1) and (2) in the Pacific (first row), Atlantic/Indian/Arctic (AIA) oceans collectively (second row), and their drainage basins.** First column: associated land drainage basins; second column: ocean basins. Labels in the vertical axis correspond to the mean ± standard deviation of the associated curve. Thick lines are the low pass filtered signal by a Hann function of 24 months. All curves in the same panel are plotted on the same scale. $P$, $E$, and $P$–$E$ are from ERA5 dataset; $dW$ is estimated from GRACE; $R$ and $N$ are estimated as a residual in equations 1 and 2, respectively.

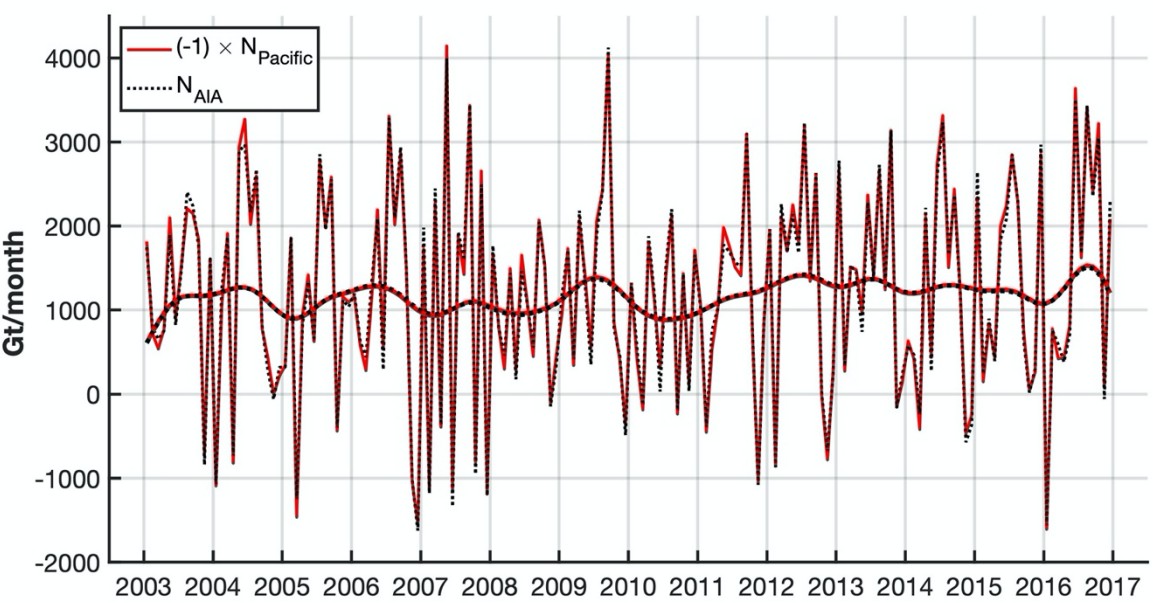

**Figure 3. Monthly time series of WT flux from the Pacific to the AIA Oceans.** Red curve is (the opposite of) the Pacific outflow, and black curve is the AIA inflow. Thick lines are the low pass filtered signal by a Hann function of 24 months.

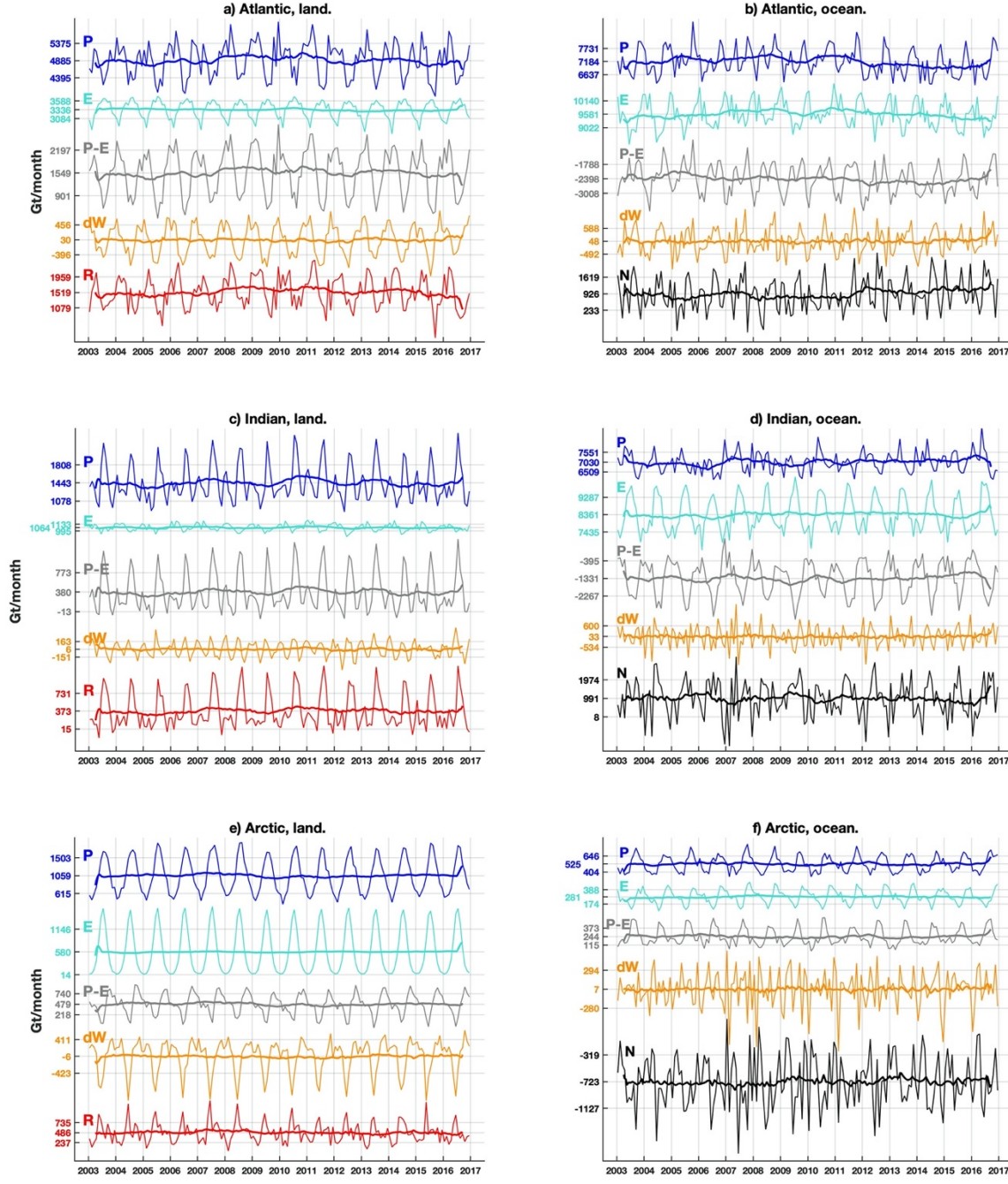

Figure 4. As Figure 2 but for Atlantic, Indian, and Arctic Oceans.



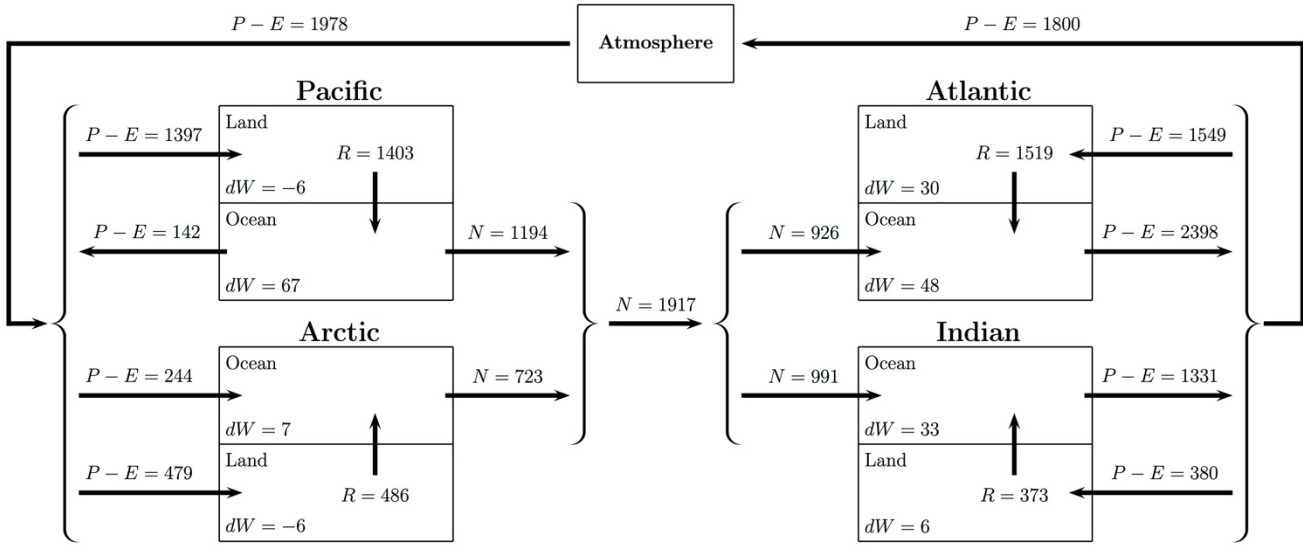

**Figure 5. Diagram of the mean values of the WT of the studied regions.** Units are Gt/month.

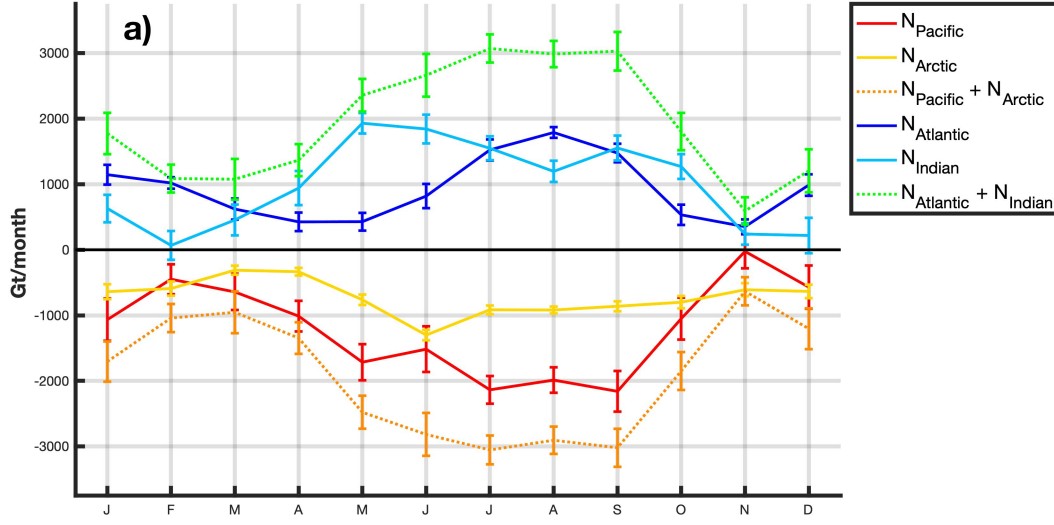


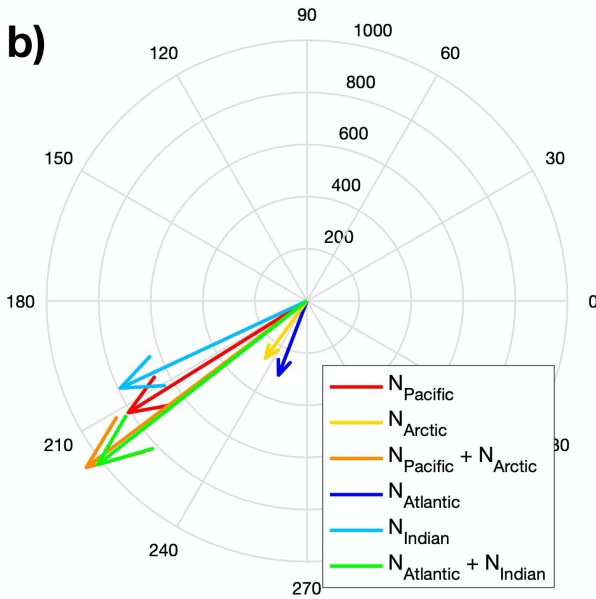

**Figure 6. (a) Annual climatology time series (error bar is one standard deviation), and (b) phasor diagram (amplitude in unit of Gt/month, phase angle according to Equation 3) of the inflow/outflow WT of the ocean basins.**

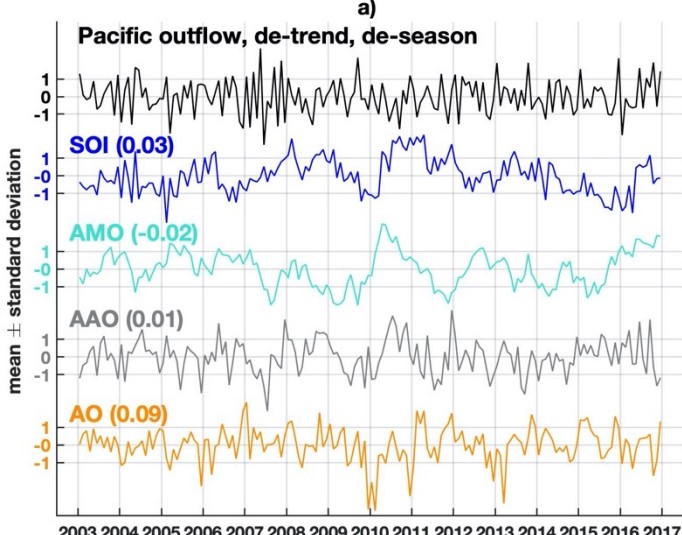


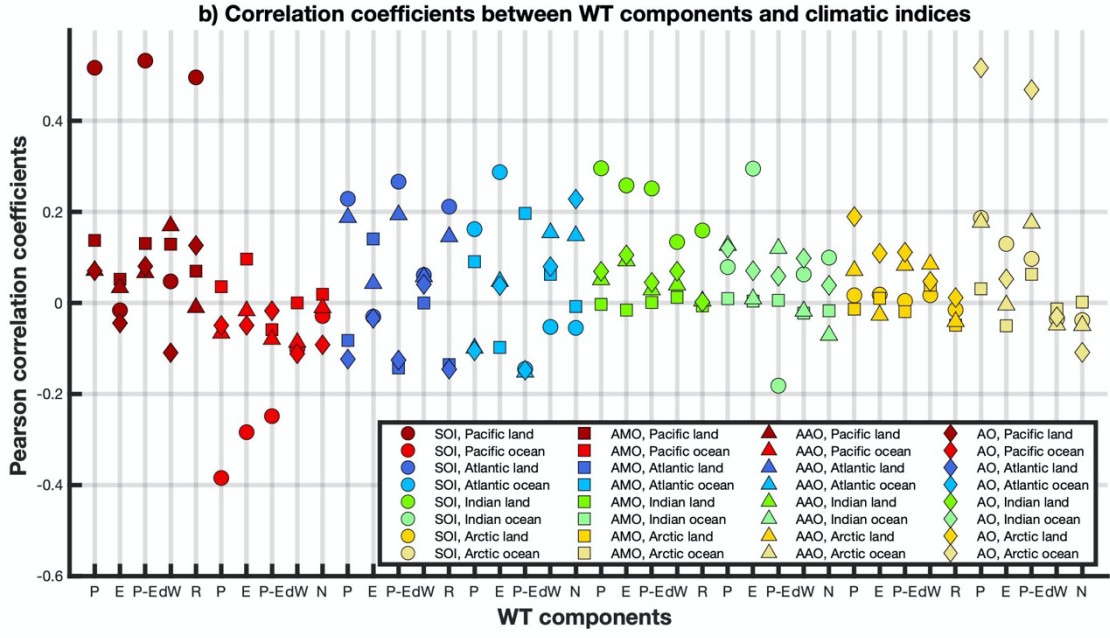

**Figure 7. Pacific outflow and climatic indices for ENSO, AMO, AO, and AAO.** a) Time series of Pacific outflow is de-trend and de-season. All time series are normalized to have unit variance. Values in the parenthesis are the correlation coefficient between the corresponding climatic index and the Pacific outflow. b) Correlation coefficients between de-trend and de-season WT components of 755 different regions and the climatic indices.

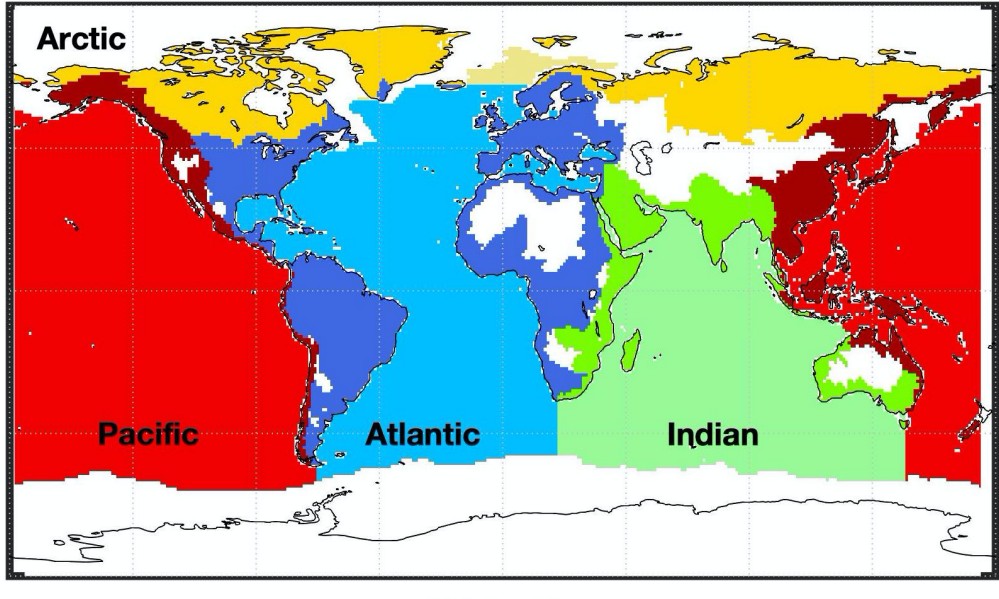

## a) Common spatial coverage to all P and E datasets

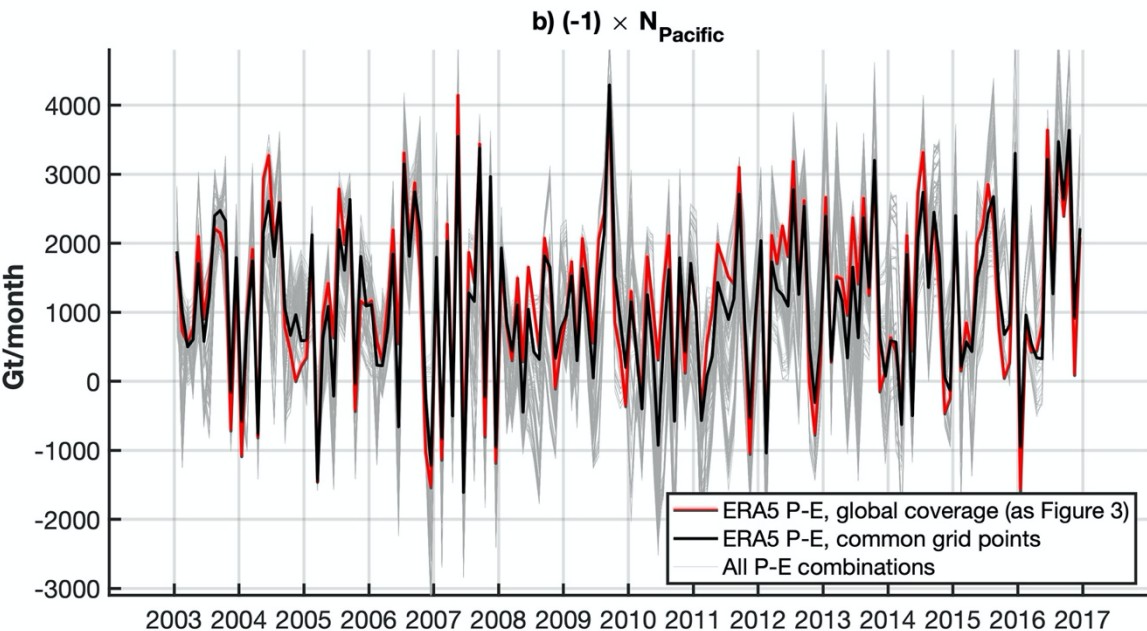

## b) (-1) × $N_{Pacific}$

**Figure 8. Monthly time series of (the opposite of) the Pacific outflow estimated from 162 combinations of *P* and *E* datasets.** a) Spatial coverage common to all datasets. b) Pacific outflows: Gray thin curves are the 162 Pacific outflows estimated in the common grid points to all datasets (no global coverage); black and red curves are based on ERA5 *P* and *E* and are obtained using either only the grid points common to all datasets (black curve) or global coverage (red curve). Note that the red curve is the same as in Figure 3.

**Table 1. Mean and annual signals of the *N* component as estimated from CSR mascon solution for different ocean basins according to Equation 2.**

|  |  | Mean (CI$_{95}$) | Annual signal (CI95) | | |
|---|---|---|---|---|---|
|  |  | (Gt/month) | Amplitude (Gt/month) | Phase (degree) | Peak date |
| Outflows | Pacific | 1194 (1087, 1308) | 809 (637, 975) | 212 (200, 224) | August 3$^{rd}$ |
|  | Arctic | 723 (709, 738) | 271 (242, 302) | 234 (228, 240) | August 25$^{th}$ |
|  | Pacific + Arctic | 1917 (1826, 2010) | 1061 (904, 1216) | 217 (209, 225) | August 8$^{th}$ |
| Inflows | AIA | 1194 (1086, 1304) | 767 (610, 926) | 212 (199, 224) | August 3$^{rd}$ |
|  | Atlantic | 926 (863, 991) | 305 (219, 384) | 249 (234, 266) | September 9$^{th}$ |
|  | Indian | 991 (911, 1067) | 791 (664, 918) | 205 (196, 214) | July 27$^{th}$ |
|  | Atlantic + Indian | 1917 (1821, 2015) | 1020 (876, 1172) | 218 (209, 226) | August 9$^{th}$ |

**Table 2. Mean of the *N* component as estimated from JPL mascon solution for different ocean basins according to Equation 2 .** $CI_{95}$ are estimated as propagation of mascon errors provided by JPL, and from bootstrap analysis. Units are Gt/month.

| | | Mean (CI$_{95}$ from error propagation) | Mean (CI$_{95}$ from bootstrap) |
|---|---|---|---|
| Outflows | Pacific | 1182 (1143, 1220) | 1182 (1062, 1306) |
| | Arctic | 735 (713, 757) | 735 (711, 761) |
| | Pacific + Arctic | 1917 (1872, 1961) | 1917 (1806, 2036) |
| Inflows | AIA | 1183 (1092, 1274) | 1183 (1077, 1282) |
| | Atlantic | 919 (866, 972) | 919 (845, 985) |
| | Indian | 999 (980, 1018) | 999 (928, 1067) |
| | Atlantic + Indian | 1918 (1862, 1974) | 1918 (1838, 2003) |

775

**Table 3. Correlation coefficients between SOI and de-trend and de-season WT components involved to estimate the Pacific outflow according to Equations 3 and 4.**

| | $std(X_i)$ (Stand. Deviation) | $corr(X_i, SOI)$ (Correlation between Xi with SOI) | $\dfrac{std(X_i)}{std(N)}$ (Coefficients) | $corr(X_i, SOI) \cdot \dfrac{std(X_i)}{std(N)}$ (Correlation ·Coefficient) |
|---|---|---|---|---|
| X1= –(P–E)ocean | 605 | 0.25 | 0.57 | 0.14 |
| X2= –(P–E)land | 212 | -0.53 | 0.20 | -0.11 |
| X3= dWland | 96 | 0.048 | 0.09 | 0.004 |
| X4= dWocean | 711 | -0.10 | 0.67 | -0.07 |
| | | | Corr(N,SOI) | -0.03 |