# Peer review of "Water transport among the world ocean basins within the water cycle"

_Earth System Dynamics, 2020_

## Referee Comment (RC1) · Anonymous Referee #1 · 8 Aug 2020

Overall this is an excellent manuscript, presenting a new result about the Earth's ocean-land-atmosphere mass exchange, using a unique combination of satellite and reanalysis datasets, and a clear easy-to-follow methodology.

The only major concern/question I have is this: the interbasin ocean transport N is a small residual of differencing large numbers. I see that each set of numbers is followed by a 95% confidence range, and I read without quite understanding that the confidence interval is computed by a bootstrap method on the data itself. I don't believe the reanalysis data have their own error estimates; I believe the GRACE data do but those did not seem to be used in the confidence interval estimation. I wonder whether estimating uncertainties in the transports by propagating uncertainties in the inputs would give intervals consistent with those of the bootstrap method. Upper bounds on the

uncertainties in the inputs can be estimated, for example, by comparing UT-CSR mascons to JPL or GSFC mascons, by comparing ECMWF reanalysis to NCEP or another model's reanalyses, etc. I say this because the lack of correlation between the inter-annual transports and ANY index of ocean-atmosphere interaction (ENSO, SOI, etc) is suspicious.

Now addressing some details: Figure 1: I would have liked to see a row with P-E-R next to the row for dW in Figure 1. Figures 1 and 3: I am sure the authors know better smoothers than the running mean (Hanning, Kaiser, etc). I recommend they use one. Line 27: Clark reference missing. Recheck all your references, I did not do an exhaustive check. Line 93: tectonic signals in the gravity field do not 'masquerader as mascons'. Mascons are a simple mathematical representation of the gravity field with a physical interpretation. Tectonics "would be incorrectly interpreted as water mass flux" Lines 124 et seq: see my concern above. A physical interpretation of this mathematical approach to confidence itnervals would be useful. Line 164: and loses 'to the atmosphere' 879 Gt/month... Line 188: I think 'The Atlantic/Arctic inflow 'mirrors this behaviour' is a better phrase in English. Somewhere: W. T. Liu et al (GRL 2006, on South American water balance) did a similar estimation of water flux between an ocean basin and the land, without using any numerical model data. There are a few more minor language errors (lines 255, 267 and possibly others). Please go over the manuscript and clean up.

---

## Referee Comment (RC2) · Munir Nayak (Referee) · 16 Aug 2020

Review of paper titled "Water transport among the world ocean basins within the water cycle" submitted to Earth System Dynamics,

Manuscript Number: esd-2020-54

Dear Editor

Thank you for sending the manuscript to me for a review. Below you will see a short summary of the manuscript, followed by my specific (major) comments, and then technical corrections (minor comments) at the end.

**Summary**

The wind-driven, heat-driven and salinity driven circulations of the oceanic waters balance the water transport among different ocean basins, atmosphere, and the continents. These transports are important in deriving climate of the earth system. In this manuscript, the authors use the GRACE observations of monthly net mass of the basins to derive the mass balance of land-atmosphere-ocean systems. The authors note that data on lateral water transport through the boundaries of oceans are sparse, and GRACE estimates can prove to be useful in estimating the mass balances. The mass balances of four oceanic basins are considered and the output variable of interest, the water mass transport/transfer among different oceanic basins, is discussed further. Among some important results, it is shown that the Pacific oceans transfers a large quantify of water to Atlantic and Indian oceans throughout the year.

The mass balance of oceanic basins supplemented by GRACE data is a very simple yet powerful idea that I think it can be of interest to many readers of ESD and other similar journals. There, however, are many concepts and ideas in the manuscript where authors need to provide

Munir Ahmad Nayak

more justifications and clarifications; I added these and many important suggestions in the "specific comments" section below. I will be looking forward to reading a revised draft.

**Specific comments**

1. I feel a strong motivation for estimating the lateral water transports from oceanic boundaries is lacking. It is not clear why we require measurements of lateral transports, given that the overall transport among different oceanic basin is known to a reasonable degree of certainty, such as by the studies noted by the authors in the introduction and discussion sections. The authors should write a concise and clear paragraph of why it is important to estimate the water fluxes through boundaries with this novel approach.

2. In addition, the authors note in their introduction that their method improves upon the previous estimates. The literature on previous estimates and how (and how much) the new method improves upon them is not discussed in detail. In addition, a critical comparison of previous estimates of water transports and the estimates provided in this study is lacking.

3. The authors observe that loss through E-P is much more in AIA as compared to the Pacific, even though the surface area is same. However, the reasons for such disparity is not discussed. Similarly, the potential reasons for other important results are not discussed. I hope to see some discussion on the results from this study.

4. Results section (L160-170). As of now, when I read the number and where the losses and gains take place, it is difficult for me to visualize the transfers among different basins. I strongly suggest the authors to present this information in terms of a multi-panel graph/map, where each map shows specific water transfer-related variable (such as N, R, etc.) with thick arrows giving the direction of transport, their color showing the magnitude of transport or we can just add text (number) inside the arrows to show magnitude.

Munir Ahmad Nayak

5. L218: This is related to my comment#3. If none of the major indices shows strong correlation with the Pacific outflow, we do not have a confidence in what causes the interannual variability of Pacific Ocean outflow. Perhaps, more detailed insights from P and E series might help and/or some literature review on this might guide the authors in understanding the likely causes of the interannual variability of the Pacific Ocean outflow. Likewise, it would be useful to perform the same analysis on other basins for understanding their interannual variability of outflows.

**Technical corrections**

L44: Correct 'de' here

Many details are missing: Such as how were E and R computed? Are these also taken from ERA5?

Figure 1: What is the source of Figure 1? Or how was this figure obtained?

Figure 1: Please add color labels to the figure. I also think the figure would look better is you add the basin names in the figure itself.; perhaps major continental basins as well.

In addition, major known pathways of water transport can be added to the figure.

L93: Elaborate more.

L94-L95: Please provide more details as to why this is required.

L98: Have you defined "GAD" earlier? Please provide details. Also, provide detailed justification of effects ignored here.

L97-100: This sentence is not very clear, rephrase for better clarity.

L99: What is MPIOM?

L105 to 115: I am not able to quite follow these sentences, perhaps because of my lack of expertise in GRACE. I'd suggest writing them in more details for readers who are not well-versed with GRACE estimates. In fact, I feel the entire paragraph can be re-written for more clearity.

Munir Ahmad Nayak

L126: Bootstrap replications of what variables? Are these timeseries data?

L128-130: Please write these in more details. "subtracted" what from what? Please clarify.

L135: All these can be written concisely in a table.

Figure 2: Should be made clearer. Shouldn't "R" be same in both left and right columns. Perhaps, you can remove the space between columns and extended the x-axis of the plots.

L145: The P-E estimate of 142 Gt/month should be corrected for final presentation, since it was earlier noted that ERA5 may have some internal inconsistency when comparing previously estimated values from the literature.

Figure 2: It might be useful to show N in right panels. Perhaps, you can only keep P- E, rather than showing both P and E.

L145: How does the results on fluxes in this manuscript compare with the values in the literature.

Figure 3: I'm not sure if Figure 3 is required. I think it can be removed without loss of any critical details.

L145 to 155: I would suggest the values to be put in a graphical format, which might look more appealing that writing the numbers.

L164: What is basis of deciding the salinity of water in this study?

L181: I though the data is monthly; if so, how can we say day of month also?

L198: Again, it seems important that authors add N time series also in Figures 2 and 4.

Figure 6: In addition to showing the timeseries, you can show the correlation values of each index with Pacific outflow.

L225: The importance of these changes has not been discussed before in the manuscript.

Appendix:

Have we referend to Appendix anywhere in the main text?

Munir Ahmad Nayak

---

## Referee Comment (RC3) · Anonymous Referee #3 · 26 Aug 2020

Review for esd-2020-54 "Water transport among the world ocean basins within the water".

In general, this study is interesting to me, with the global ocean basins to study water mass transport based on GRACE and ERA5. The conclusions are generally supported by the data analyses in this study, but more validations/evaluations are necessary to improve the reliability of the results. At least, a careful inter-comparsion between this study and previous studies/literatures can be discussed to enhance our understanding.

Other comments:

L44, change 'de' to 'the'.

In section 2: Methodology and Data: please use subtitles to re-organize the section,

[Figure]

and improve the readability.

Fig 1: suggest to add legends to indicate the locations of different ocean basins.

Fig 2: are these values obtained from GRACE? or a combination of ERA5 and GRACE? please briefly clarify this in the figure caption.

Fig 3: black curve is the AIA inflow: 'if' should be 'is'.

Fig 4: N component should be briefly explained in the caption.

Fig 6: please indicate the correlation coefficient R between each climate index and the Pacific outflow, at each time series.

————————————————————

---

## Author Comment (AC1) · 3 Sep 2020

**Answer to Reviewer 1**

We thank the reviewer for thorough reading and thoughtful comments and suggestions. A detailed discussion of the changes that we made in response to the reviewer's comments is given below. In what follows, we state the reviewer's comment in boldface, and describe our response in plain text. Text in the manuscript is represented in italics. The text that has been modified/included in the new version has been highlighted in red.

**"Overall this is an excellent manuscript, presenting a new result about the Earth's ocean-land-atmosphere mass exchange, using a unique combination of satellite and reanalysis datasets, and a clear easy-to-follow methodology."**

We appreciate the positive overall comment about the manuscript.

**"The only major concern/question I have is this: the interbasin ocean transport N is a small residual of differencing large numbers. I see that each set of numbers is followed by a 95% confidence range, and I read without quite understanding that the confidence interval is computed by a bootstrap method on the data itself. I don't believe the re-analysis data have their own error estimates; I believe the GRACE data do but those did not seem to be used in the confidence interval estimation. I wonder whether estimating uncertainties in the transports by propagating uncertainties in the inputs would give intervals consistent with those of the bootstrap method. Upper bounds on the uncertainties in the inputs can be estimated, for example, by comparing UT-CSR mascons to JPL or GSFC mascons, by comparing ECMWF reanalysis to NCEP or another model's reanalyses, etc. I say this because the lack of correlation between the inter- annual transports and ANY index of ocean-atmosphere interaction (ENSO, SOI, etc) is suspicious."**

The following changes have been included to address the issues raised by the referee:

1. **Bootstrap:** We have included an intuitive description of the bootstrap method for time series and a reference to a paper on bootstrap method for time series. Besides, we have provided extended details about how confidence intervals have been evaluated:

   *The reported 95% confidence intervals and the correlation coefficients are evaluated using the stationary bootstrap scheme of Politis and Romano (1994) (with optimal block length selected according to Patton et al., 2009), and the percentile method. The intuition underlying the bootstrap is simple. Suppose that the observed time series $x_1$, ..., $x_n$ is a realization of the random vector $(X_1,..., X_n)$ with joint distribution $P_n$ and which is assumed to be part of a stationary stochastic process. Given $\boldsymbol{X}_n$, we first build and estimate $\hat{P}_n$ of $P_n$. Then B random vectors $(X_1^*, ..., X_n^*)$ are generated from $\hat{P}_n$. If $\hat{P}_n$ is a good approximation of $P_n$, then the relation between $(X_1^*, ..., X_n^*)$ and $\hat{P}_n$ should well reproduce the relation between $(X_1,..., X_n)$ and $P_n$ (for an introduction of bootstrap methods for time series see Kreiss and Lahiri (2012) and the references therein). Here, the number of bootstrap replications was set to B=2000. In general, half length of the confidence interval can be very well approximated by twice the standard deviation of the sample mean estimated from the bootstrap replications. Prior to applying the bootstrap to a time series, least-squares estimated linear/quadratic trend and sinusoid with the most relevant frequencies are removed from it to meet the stationarity conditions of the method. In particular, each series*

*has been decomposed into trend, seasonal and residual components. The bootstrap is applied to the residual component producing bootstrap samples of the residuals. For the evaluation of confidence intervals for the different components of WT, the trend and seasonal terms are added back (to the bootstrap sample of the residuals) producing bootstrapped time series of the component of interest. These samples are then used for further analysis. As an illustration,* for the WT N component we proceed as follows: (i) a model with linear, annual, and semiannual signals is fitted to the data. *The fitted linear trend and annual and semiannual signals are subtracted from the original time series; (ii) the stationary bootstrap is then applied to the residuals producing 2000 bootstrap samples of the residuals; (iii) The estimated trend and seasonal components are added back to each bootstrap sample of the residuals obtaining an ensemble of 2000 bootstrapped time series for the N component; (iv) these 2000 bootstrapped time series are used to obtain 95% confidence intervals for the mean fluxes (average of N over the 14 year period of study) and for the amplitude and phase of the annual component using the percentile method. For the mean fluxes, the average of N for each of the 2000 bootstrapped time series was first evaluated and then the 0.025 and 0.975 percentiles of these 2000 averages were reported as 95% confidence interval. For the study of the climatology, a linear trend model with annual and semiannual components was fitted to the 2000 bootstrapped time series producing corresponding estimates of the annual amplitude and phase. The 0.025 and 0.975 percentiles of these estimates were reported as 95% confidence intervals.* In order to study the robustness of the results with respect to the model choice, the analysis is rerun using 11 alternative models obtained considering different forms for the trend component (quadratic or constant) and including higher frequencies in the harmonic regression (up to 5). The results are robust. The relative difference with respect to the reported values is smaller than 1.2% for point estimates and smaller than 3.3% for the extremes of the 95% confidence intervals.

2. **Confidence intervals of the correlation coefficients**. More details are provided:

   *Note that for the study of correlation the bootstrap was applied to the bivariate time series of the residuals of the two variables of interest producing an ensemble of 2000 bivariate time series of residuals. For each bivariate time series of residuals the correlation between the two components of the series was first evaluated. The average and the 0.025 and 0.975 percentiles of these 2000 estimates were reported as point estimate and confidence limits for the correlation between the two variables of interest (correlation between residual components is used to avoid spurious correlation).*

3. **Bootstrap Vs Error propagation:** The confidence intervals estimated from bootstrap have been compared to those estimated from error propagation of the mascon. As CSR mascon solution does not provide such error estimates, we have used the JPL mascon solution for the comparison. An explanation of why bootstrap confidence intervals contains, as expected, the error propagation confidence interval has been also provided. In the description of the bootstrap method we have included the following text:

   *As an independent check of the bootstrap, confidence intervals for the mean value of N have been also evaluated by propagating the error estimate in GRACE data (using the JPL GRACE mascon solution for which error estimates are available). The resulting intervals were*

*consistent with those of the bootstrap method. In particular (see Section 4 for details), we show that in all cases the bootstrap intervals contain the intervals obtained from error propagation. In this respect, the $CI_{95}$ from bootstrap analysis can be considered a conservative estimate. This should be expected, since the residual component underlying the bootstrap approach includes measurement errors and other type of errors (related, for example, with the estimate of the trend and seasonal terms). As a result, the uncertainties in the transports estimated by the bootstrap should be larger than the corresponding uncertainties estimated by error propagation.*

We have included a new section 4, entitled "Comparison with other datasets", which includes the comparison between error propagation and bootstrap confidence intervals for the *N* component estimated from JPL data:

*CSR GRACE mascon solution is replaced by the JPL GRACE mascon solution provided by the Jet Propulsion Laboratory/NASA (Watkins et al., 2015; Wiese et al., 2019). Similarly to CSR data, JPL are corrected for GIA effects, $C_{20}$ Stoke coefficients are replaced by a solution from SLR, and data are reduced to 1º regular grids from 0.5º regular grids. Besides, we have applied the degree-0 Stoke coefficients correction. However, CSR and JPL mascon solutions are not directly comparable. The main reason is that an estimate of degree-1 coefficients has been added to JPL mascon solutions, and the GAD product has not been added back. The corrections applied by JPL are not supplied separately and we cannot do/undo any of the corrections to process JPL data as we did with CSR data. In particular, the GAD product is not available for JPL. In any case, the JPL solution is useful here since it provides an error estimate of the mascon solution that can be propagated to obtain confidence intervals of N, which are independent from those estimated with the bootstrap analysis. Table 2 shows the $CI_{95}$ of the mean values of the N component for different ocean basin estimated from error propagation and bootstrap analysis. It is observed that in all cases the $CI_{95}$ from error propagation are included in those from bootstrap analysis, meaning that the latter are a conservative estimate of the error. JPL propagated error can be expected to be similar to that propagated from CSR error estimates (which are not available), and then we can assume that the reported $CI_{95}$ for N calculated from CSR data are a conservative estimate. Besides, comparing Tables 1 and 2, it is observed that the mean values of N are quite similar and that the $CI_{95}$ largely overlap. Regarding to the time variability, the values of the N component from CSR and JPL mascon solutions show Pearson correlation coefficients greater than 0.85 (p-value < $10^{-3}$), except for the Atlantic (0.70). Thus, despite the different processing of CSR and JPL data, the reported analysis for the N component is robust with respect to the choice of GRACE datasets.*

[revised manuscript text omitted]

5. **Lack of correlation.** We have included a discussion on the lack of correlation between the inter-annual transports and the indices of ocean-atmosphere interaction. In particular we propose the two following explanations:

   *"To explore this lack of correlation, we have estimated the correlation coefficient between each climatic index and each WT component (Figure 7b).*

[Figure]

[Figure]

**Figure 7. Pacific outflow and climatic indices for ENSO, AMO, AO, and AAO. a)** Time series of Pacific outflow is de-trend and de-season. All time series are normalized to have unit variance. Values in the parenthesis are the correlation coefficient between the corresponding climatic index and the Pacific outflow. b) Correlation coefficients between de-trend and de-season WT components of different regions and the climatic indices.

*All of them are lower than 0.3 except for 6 cases in 2 regions. In the Arctic, P and P−E in the drainage basins of the Arctic show a correlation of ~0.5 with the AO. This correlation is natural since that is the area of influence of the AO. The other region is the Pacific, where, as expected, the SOI shows a correlation around 0.5 with P, P−E, and R in the drainage basins, and around −0.4 with P in the ocean. However, this individual correlation does not extend to the Pacific outflow. In order to understand why this is the case, it is convenient to express the N component of the water transport as a function of (P-E) and dW. According to Equations 1 and 2 we have:*

$$N = -(P-E)_{ocean} - R + dW_{ocean} = \\ \underbrace{-(P-E)_{ocean}}_{X_1} \underbrace{-(P-E)_{land}}_{X_2} + \underbrace{dW_{land}}_{X_3} + \underbrace{dW_{ocean}}_{X_4} \qquad (3)$$

*It can be shown that the correlation between N and a given index can be express as follows*

$$corr(N, Index) = \sum_{i=1}^{4} corr(X_i, Index) \cdot \frac{std(X_i)}{std(N)}, \qquad\qquad (4)$$

*where corr denotes the correlation coefficient, and std stands for standard deviation. As shown in equation (4), the correlation between N and a given index is a linear combination of the correlation between each component and the index. The coefficients of the linear combination std($X_i$)/std(N) are proportional to the standard deviation of each component. The components of equation (4) for the Pacific outflow and the SOI index are shown in Table 3. Despite the fact that some of the individual component exhibits significant correlation with SOI (in particular P–E in land and ocean) when combined with the corresponding coefficients their effects are canceled out yielding to a negligible correlation between water transport and SOI (below 0.03 in magnitude).*

*Another possible reason for the lack of correlation resides in the definition of the studied regions, for which the presence of subregions with positive and negative influence of an index results in an overall negligible/attenuated influence of the index in the overall region. For example, a positive phase of the AMO is related to an increase of P in western Europe (Sutton and Hodson, 2005), and the Sahel (Folland et al., 1986; Knight et al., 2006; Zhang and Delworth, 2006; Ting et al., 2009), but to a decrease of P in the U.S. (Enfield et al., 2001; Sutton and Hodson, 2005), and northeast Brazil (Knight et al., 2006; Zhang and Delworth, 2006). All these regions are included in the Atlantic drainage basin, and then the influence of a positive phase of the AMO is attenuated."*

**Table 3. Correlation coefficients between SOI and de-trend and de-season WT components involved to estimate the Pacific outflow according to Equations 3 and 4.**

| | $std(X_i)$ (Stand. Deviation) | $corr(X_i, SOI)$ (Correlation between $X_i$ with SOI) | $\dfrac{std(X_i)}{std(N)}$ (Coefficients) | $corr(X_i, SOI) \cdot \dfrac{std(X_i)}{std(N)}$ (Correlation ·Coefficient) |
|---|---|---|---|---|
| $X_1 = -(P–E)_{ocean}$ | 605 | 0.25 | 0.57 | 0.14 |
| $X_2 = -(P–E)_{land}$ | 212 | -0.53 | 0.20 | -0.11 |
| $X_3 = dW_{land}$ | 96 | 0.048 | 0.09 | 0.004 |
| $X_4 = dW_{ocean}$ | 711 | -0.10 | 0.67 | -0.07 |
| | | Corr(*N*,SOI) | | -0.03 |

Note that table 3 provides also some insights about the causes of the interannual variability of Pacific Ocean outflow. The largest standard deviation of $P–E$ and $dW$ in the ocean suggests that these two components might drive the interannual variability of the Pacific Ocean outflow. This is confirmed by a correlation analysis. The correlation between $N$ and the $(P–E)_{Ocean}$ is -0.70. The correlation between $N$ and the $dW_{ocean}$ is 0.84. The correlation of $N$ with the corresponding land components is below 0.18. In all cases, prior to the evaluation of the correlation the corresponding time series have been de-trend and de-season.

**Now addressing some details:**

**(1) Figure 1: I would have liked to see a row with P-E- R next to the row for dW in Figure 1.**

Figure 1 probably means Figure 2. Including *P-E-R*, in our opinion, is not very useful since, by definition of *R*, *P-E-R* will perfectly match *dW*. The comparison would be interesting with an independent dataset of R.

**(2) Figures 1 and 3: I am sure the authors know better smoothers than the running mean (Hanning, Kaiser, etc). I recommend they use one.**

We have replaced the running mean by a low pass filter defined by a Hann function of 24 months (the resulting smoothed curve is quite in agreement with the one previously obtained by running mean smoothing)

**(3) Line 27: Clark reference missing. Recheck all your references, I did not do an exhaustive check.**

Thank you. We have checked all the references.

**(4) Line 93: tectonic signals in the gravity field do not 'masquerader as mascons'. Mascons are a simple mathematical representation of the gravity field with a physical interpretation. Tectonics "would be incorrectly interpreted as water mass flux"**

Thank you. It is better expressed in this way. We have re-written the sentence: *"Any other non-surficial effect such as long-term tectonics would be incorrectly interpreted as water mass fluxes…"*

**(5) Lines 124 et seq: see my concern above. A physical interpretation of this mathematical approach to confidence intervals would be useful.**

We have extended the description of the bootstrap - see point 3 (Bootstrap Vs Error propagation) in page 1 of this response.

**(6) Line 164: and loses 'to the atmosphere' 879 Gt/month. . .**

The sentence has been re-written:
*"On average, the Atlantic Ocean receives 926 Gt/month ($CI_{95}$=[876, 980]; or 0.36 Sv) of salty water, and loses to the atmosphere 879 Gt/month ($CI_{95}$=[828, 930]) via P−E+R."*

**(7) Line 188: I think 'The Atlantic/Arctic inflow 'mirrors this behaviour' is a better phrase in English.**

Thank you. We have re-written the sentence: *"The Atlantic/Arctic inflow mirrors this behaviour."*

**(8) Somewhere: W. T. Liu et al (GRL 2006, on South American water balance) did a similar estimation of water flux between an ocean basin and the land, without using any numerical model data.**

Thank you. We agree that it is a pertinent reference. We have included it in the last paragraph of the introduction, which now is:

*"In this work we propose a new methodology devised to estimate the net WT through the boundaries of a given oceanic region. A defining feature of the proposed approach is the use of the time-variable gravity data from the GRACE (Gravity Recovery and Climate Experiment) satellite mission to estimate* the *change of water content. We apply the methodology, in conjunction with conventional meteorological data of general hydrologic budget schemes, to estimate the time evolution over the period 2003-2016 of the net WT and exchanges among the four major ocean basins – namely Pacific, Atlantic, Indian, and Arctic. We analyse and report our results of the seasonal climatology as well as the interannual variability of WT. Such information, not available previously, is of valuable importance. For example, in closed regions, net WT through the boundaries on the surface must be counteracted by moisture fluxes through the same boundaries in the atmosphere to match GRACE measurements. Such approach has been successfully applied to study the hydrological cycle of South America (Liu et al., 2006). At ocean basin scale, knowledge about net WT not only would help elucidate the role of the oceans within the water cycle, but it will also impose restrictions on moisture advection in the atmosphere that would help to improve atmospheric models. On the other hand, ocean models usually deal with inflows and outflows of a given ocean region (Warren, 1983; Rahmstorf, 1996; Emile-Geay et al., 2003; de Vries and Weber, 2005; Dijkstra, 2007). Net WT estimates for such ocean region would be useful to impose constraints to the relationship between its inflows and outflows, which would improve the reliability of the models. Better models will improve our knowledge of the Earth's WT dynamics and its evolution in the future, which is critical in the present scenario of climate change."*

**(9) There are a few more minor language errors (lines 255, 267 and possibly others). Please go over the manuscript and clean up.**

Done. Thank you.

---

## Author Comment (AC2) · 3 Sep 2020

**Answer to Reviewer 2**

We thank the reviewer for thorough reading and thoughtful comments and suggestions. A detailed discussion of the changes that we made in response to the reviewer's comments is given below. In what follows, we state the reviewer's comment in boldface, and describe our response in plain text. Text in the manuscript is represented in italics. The text that has been modified/included in the new version has been highlighted in red.

**Specific comment 1:**

**I feel a strong motivation for estimating the lateral water transports from oceanic boundaries is lacking. It is not clear why we require measurements of lateral transports, given that the overall transport among different oceanic basin is known to a reasonable degree of certainty, such as by the studies noted by the authors in the introduction and discussion sections. The authors should write a concise and clear paragraph of why it is important to estimate the water fluxes through boundaries with this novel approach.**

The last paragraph of the introduction has been re-written:

*"In this work we propose a new methodology devised to estimate the net WT through the boundaries of a given oceanic region. A defining feature of the proposed approach is the use of the time-variable gravity data from the GRACE (Gravity Recovery and Climate Experiment) satellite mission to estimate the change of water content. We apply the methodology, in conjunction with conventional meteorological data of general hydrologic budget schemes, to estimate the time evolution over the period 2003-2016 of the net WT and exchanges among the four major ocean basins – namely Pacific, Atlantic, Indian, and Arctic. We analyse and report our results of the seasonal climatology as well as the interannual variability of WT. Such information, not available previously, is of valuable importance. For example, in closed regions, net WT through the boundaries on the surface must be counteracted by moisture fluxes through the same boundaries in the atmosphere to match GRACE measurements. Such approach has been successfully applied to study the hydrological cycle of South America (Liu et al., 2006). At ocean basin scale, knowledge about net WT not only would help elucidate the role of the oceans within the water cycle, but it will also impose restrictions on moisture advection in the atmosphere that would help to improve atmospheric models. On the other hand, ocean models usually deal with inflows and outflows of a given ocean region (Warren, 1983; Rahmstorf, 1996; Emile-Geay et al., 2003; de Vries and Weber, 2005; Dijkstra, 2007). Net WT estimates for such ocean region would be useful to impose constraints to the relationship between its inflows and outflows, which would improve the reliability of the models. Better models will improve our knowledge of the Earth's WT dynamics and its evolution in the future, which is critical in the present scenario of climate change."*

**Specific comment 2:**

**In addition, the authors note in their introduction that their method improves upon the previous estimates. The literature on previous estimates and how (and how much) the new method improves upon them is not discussed in detail. In addition, a critical comparison of previous estimates of water transports and the estimates provided in this study is lacking.**

As far as we are applying a new methodology, there are not many studies to compare with. The only two studies, up to our knowledge, doing something similar are discussed in the third paragraph of the section "Discussion and Conclusions":

*"The results presented here are consistent with the well-known salinity asymmetry between the Pacific and Atlantic Oceans (Reid, 1953; Warren, 1983; Broecker et al., 1985; Zaucker et al., 1994; Rahmstorf, 1996; Emile-Geay et al., 2003; Lagerloef et al., 2008; Czaja, 2009; Reul, 2014). However, they are in contrast to previous GRACE-based studies where a simple seesaw WT between the Pacific and the Atlantic/Indian oceans was reported (Chambers and Willis, 2009; Wouters et al., 2014). In those studies, the P−E+R term in Equation 2 in both Pacific and Atlantic/Indian Oceans was approximated by that from the global ocean mean. However, the mean freshwater flux in the Pacific (1261 Gt/month) quite mis-matches that in the Atlantic/Indian Oceans (−1837 Gt/month), meaning that the approximation was quite poor and hence the N term was not properly estimated in these studies (see Appendix for further discussion)."*

As stated in the text, in the Appendix we explain in detail why the proposed methodology overcomes some important limitations of previous approaches which will always show a seesaw of water transport, even if it does not exist.

**Specific comment 3:**

**The authors observe that loss through E-P is much more in AIA as compared to the Pacific, even though the surface area is same. However, the reasons for such disparity is not discussed. Similarly, the potential reasons for other important results are not discussed. I hope to see some discussion on the results from this study.**

The $P$, $E$, $P\text{-}E$, and $R$ components are auxiliar in this study. However, we understand the reviewer's concern and we have added some more references for comparison purposes in the last paragraph of Section 3.1:

*"Corresponding analyses have been performed for the Atlantic, Indian, and Arctic Oceans separately. The time evolution of the WT components in Eqs. 1 and 2 are shown in Figure 4, and a diagram of the water-mass fluxes is shown in Figure 5. On average, the Atlantic Ocean receives 926 Gt/month (CI$_{95}$=[876, 980]; or 0.36 Sv) of salty water, and loses to the atmosphere 879 Gt/month (CI$_{95}$=[828, 930]) via P−E+R. The latter is equivalent to a freshwater deficit of 0.34 Sv, which increases the near-surface salt concentration and enables water to sink in North Atlantic producing deep water. These values are close to the 0.13-0.32 Sv estimated from ocean models, as needed to keep*

*salinity balance in the Atlantic Ocean (Zaucker et al., 1994). Similarly, the Indian Ocean loses 957 Gt/month ($CI_{95}$=[894, 1022]) of freshwater that is restored by 991 Gt/month ($CI_{95}$=[907, 1073]) of salty water. The freshwater lost via P−E+R by the Atlantic and Indian Oceans goes to the Pacific (1261 Gt/month, $CI_{95}$=[1171, 1347]) and Arctic (730 Gt/month, $CI_{95}$=[712, 747]) Oceans, which return 1194 ($CI_{95}$=[1096, 1291]) and 723 ($CI_{95}$=[708, 739]) Gt/month of salty water through the ocean, respectively. Then, the Pacific presents a surplus of freshwater that reduces near-surface salt concentration, which prevents the formation of deep water. Together, the Pacific and Arctic Oceans supply 1917 Gt/month ($CI_{95}$=[1812, 2021]) of water to the Atlantic and Indian Oceans, where it is reincorporated into the water cycle via net E−P. As in previous studies (see Craig et al., 2017 for a synthesis), the freshwater lost in the Indian Ocean is similar to that in the Atlantic Ocean. In those studies, P−E+R is close to zero in the Pacific Ocean, producing a difference of 0.4 Sv between Atlantic and Pacific Oceans. In this study, P−E+R is 1261 Gt/month in the Pacific Ocean and the difference with the Atlantic increases to ~0.8 Sv. Some of these differences would be expected as far as the ocean basins are not defined in exactly the same way. On the other hand, the global R is 3781 Gt/month (or $3781 \times 12$ = 45368 $km^3$/year), close to the 41867 $km^3$/year reported by the Global Runoff Data Centre (GRDC, 2014). At basin scale, R is 16834 $km^3$/year in the Pacific, greater than the 11826 $km^3$/year reported by GRDC. In the Atlantic, Indian, and Arctic, R is 18228, 4479, and 5827 $km^3$/year, respectively, which is closer to the GRDC values: 20772, 5238, and 4080 $km^3$/year. Finally, according to the diagram in Figure 5, the water content in the atmosphere decreases 178 Gt/month (and it is gained by Earth's surface), but this amount is not realistic as discussed in Section 2 since it should increase a few Gt/month (Nilsson and Elgered, 2008). This value differs from the 188 Gt/month mentioned in Section 2 because the endorheic regions are not accounted here.*"

More importantly, we have extended our analysis to other datasets. The objective is to show that our main results concerning the *N component*, are not an artifact of CSR GRACE and ERA5 datasets. As a result, there is new section entitled "Comparison with other datasets":

*"Equations 1 and 2 are applied to estimate the Pacific outflow using different datasets:*

*(1) CSR GRACE mascon solution is replaced by the JPL GRACE mascon solution provided by the Jet Propulsion Laboratory/NASA (Watkins et al., 2015; Wiese et al., 2019). Similarly to CSR data, JPL are corrected for GIA effects, $C_{20}$ Stoke coefficients are replaced by a solution from SLR, and data are reduced to 1º regular grids from 0.5º regular grids. Besides, we have applied the degree-0 Stoke coefficients correction. However, CSR and JPL mascon solutions are not directly comparable. The main reason is that an estimate of degree-1 coefficients has been added to JPL mascon solutions, and the GAD product has not been added back. The corrections applied by JPL are not supplied separately and we cannot do/undo any of the corrections to process JPL data as we did with CSR data. In particular, the GAD product is not available for JPL. In any case, the JPL solution is useful here since it provides an error estimate of the mascon solution that can be propagated to obtain confidence intervals of N, which are independent from those estimated with the bootstrap analysis. Table 2 shows the $CI_{95}$ of the mean values of the N component for different ocean basin estimated from error propagation and bootstrap analysis. It is observed that in all cases the $CI_{95}$ from error*

[revised manuscript text omitted]

**Specific comment 4:**

**Results section (L160-170). As of now, when I read the number and where the losses and gains take place, it is difficult for me to visualize the transfers among different basins. I strongly suggest the authors to present this information in terms of a multi-panel graph/map, where each map shows specific water transfer-related variable (such as N, R, etc.) with thick arrows giving the direction of transport, their color showing the magnitude of transport or we can just add text (number) inside the arrows to show magnitude.**

Following the suggestion of the reviewer, we have included a new Figure with a diagram of the mean WT components to ease the reading:

[Figure]

**Figure 5. Diagram of the mean values of the WT of the studied regions.** Units are Gt/month.

**Specific comment 5:**

**L218: This is related to my comment#3. If none of the major indices shows strong correlation with the Pacific outflow, we do not have a confidence in what causes the interannual variability of Pacific Ocean outflow. Perhaps, more detailed insights from P and E series might help and/or some literature review on this might guide the authors in understanding the likely causes of the interannual variability of the Pacific Ocean outflow. Likewise, it would be useful to perform the same analysis on other basins for understanding their interannual variability of outflows.**

Following the suggestion of the reviewer we have looked in more details at the $P$ and $E$ series. This has provided some insight in both the lack of the correlation of the Pacific outflow with the most important climatic indices and the interannual variability of the Pacific Ocean outflow. In particular:

We have extended the analysis about the lack of correlation and we give two possible explanations:

*" To explore this lack of correlation, we have estimated the correlation coefficient between each climatic index and each WT component (Figure 7b).*

*All of them are lower than 0.3 except for 6 cases in 2 regions. In the Arctic, P and P−E in the drainage basins of the Arctic show a correlation of ~0.5 with the AO. This correlation is natural since that is the area of influence of the AO. The other region is the Pacific, where, as expected, the SOI shows a correlation around 0.5 with P, P−E, and R in the drainage basins, and around -0.4 with P in the ocean. However, this individual correlation does not extend to the Pacific outflow. In order to understand why this is the case, it is convenient to express the N component of the water transport as a function of (P-E) and dW. According to Equations 1 and 2 we have:*

$$N = - (P - E)_{ocean} - R + dW_{ocean} = \underbrace{- (P - E)_{ocean}}_{X_1} \underbrace{- (P - E)_{land}}_{X_2} + \underbrace{dW_{land}}_{X_3} + \underbrace{dW_{ocean}}_{X_4}. \qquad (3)$$

*It can be shown that the correlation between N and a given index can be express as follows*

$$corr(N, Index) = \sum_{i=1}^{4} corr(X_i, Index) \cdot \frac{std(X_i)}{std(N)}, \qquad (4)$$

*where corr denotes the correlation coefficient, and std stands for standard deviation. As shown in equation (4), the correlation between N and a given index is a linear combination of the correlation between each component and the index. The coefficients of the linear combination std(Xi)/std(N) are proportional to the standard deviation of each component. The components of equation (4) for the Pacific outflow and the SOI index are shown in Table 3. Despite the fact that some of the individual component exhibits significant correlation with SOI (in particular P-E in land and ocean) when combined with the corresponding coefficients their effects are cancelled out yielding to a negligible correlation between water transport and SOI (below 0.03 in magnitude).*

[Figure]

[Figure]

**Figure 7. Pacific outflow and climatic indices for ENSO, AMO, AO, and AAO. a)** Time series of Pacific outflow is de-trend and de-season. All time series are normalized to have unit variance. Values in the parenthesis are the correlation coefficient between the corresponding climatic index and the Pacific outflow. b) Correlation coefficients between de-trend and de-season WT components of different regions and the climatic indices.

*Another possible reason for the lack of correlation resides in the definition of the studied regions, for which the presence of subregions with positive and negative influence of an index results in an overall negligible/attenuated influence of the index in the overall region. For example, a positive phase of the AMO is related to an increase of P in western Europe (Sutton and Hodson, 2005), and the Sahel (Folland et al., 1986; Knight et al., 2006; Zhang and Delworth, 2006; Ting et al., 2009), but to a decrease of P in the U.S. (Enfield et al., 2001; Sutton and Hodson, 2005), and northeast Brazil (Knight et al., 2006; Zhang and Delworth, 2006). All these regions are included in the Atlantic drainage basin, and then the influence of a positive phase of the AMO is attenuated."*

**Table 3. Correlation coefficients between SOI and de-trend and de-season WT components involved to estimate the Pacific outflow according to Equations 3 and 4.**

| | $std(X_i)$

 (Stand. Deviation) | $corr(X_i, SOI)$

 (Correlation between $X_i$ with SOI) | $\dfrac{std(X_i)}{std(N)}$

 (Coefficients) | $corr(X_i, SOI) \cdot \dfrac{std(X_i)}{std(N)}$

 (Correlation ·Coefficient) |
|---|---|---|---|---|
| $X_1 = -(P{-}E)_{ocean}$ | 605 | 0.25 | 0.57 | 0.14 |
| $X_2 = -(P{-}E)_{land}$ | 212 | -0.53 | 0.20 | -0.11 |
| $X_3 = dW_{land}$ | 96 | 0.048 | 0.09 | 0.004 |
| $X_4 = dW_{ocean}$ | 711 | -0.10 | 0.67 | -0.07 |
| | | | Corr($N$,SOI) | -0.03 |

Note that table 3 provides also some insights about the causes of the interannual variability of Pacific Ocean outflow. The largest standard deviation of $P{-}E$ and $dW$ in the ocean suggests that these two components might drive the interannual variability of the Pacific Ocean outflow. This is confirmed by a correlation analysis. The correlation between $N$ and the $(P{-}E)_{ocean}$ is -0.70. The correlation between $N$ and the $dW_{ocean}$ is 0.84. The correlation of $N$ with the corresponding land components is below 0.18. In all cases, prior to the evaluation of the correlation the corresponding time series have been de-trend and de-season.

**Technical corrections**

**L44: Correct 'de' here**

Done, thank you.

**Many details are missing: Such as how were E and R computed? Are these also taken from ERA5?**

We have clarified that $P$ and $E$ data are both from ERA5 at the beginning of the description of ERA5 dataset:

*"The P and E data we use are adopted from the ERA5 reanalysis…"*

We have also clarified that $R$ is estimated as a residual in Equation 1 (in the description of Equatin 1):

*"The R component will be estimated as a residual in Equation 1."*

**Figure 1: What is the source of Figure 1? Or how was this figure obtained?**

The source of Figure 1 is the runoff pathways scheme of Oki and Sud (1998) as stated in lines 75-76 of the original manuscript:

*"The land is divided into their associated drainages according to the global continental runoff pathways scheme of Oki and Sud (1998)"*

To avoid any confusion, we added in the caption of Figure 1 the reference to runoff pathways scheme of Oki and Sud (1998). The new caption of Figure 1 is:

*Figure 1. Pacific, Atlantic, Indian, and Arctic Ocean basins and their associated continental drainage basins* *according to the global continental runoff pathways scheme of Oki and Sud (1998). Within each basin, darker colour represents the continental basin, lighter colour the ocean basin. White regions represent endorheic basins.*

**Figure 1: Please add color labels to the figure. I also think the figure would look better is you add the basin names in the figure itself.; perhaps major continental basins as well. In addition, major known pathways of water transport can be added to the figure.**

The names of the basins have been added to the figure, and in figure caption has been modified.
A diagram of net water transport has been included as a new figure 5 (see answer to specific comment 4). The goal of figure 1 is just to show how the regions have been defined. For this reason, we believe that additional partitions (as major continental basins) might be distracting. Although we agree that showing major known pathways of water transport could enrich the figure we decided not to include it to avoid confusion with the reported net water transport, which is the target of this study.

The results of the changes described above is the new Figure 1

[Figure]

*Figure 1. Pacific, Atlantic, Indian, and Arctic Ocean basins and their associated continental drainage basins* *according to the global continental runoff pathways scheme of Oki and Sud (1998). Within each basin, darker colour represents the continental basin, lighter colour the ocean basin. White regions represent endorheic basins*

**L93: Elaborate more.**

The sentence has been re-written:

*"Any other non-surficial effect such as long-term tectonics would be incorrectly interpreted as water mass fluxes (Chao, 2016) but they may only have importance in the determination of secular trends; so are the non-climatic sources such as the rare, local earthquake events."*

**L94-L95: Please provide more details as to why this is required.**

The sentence has been re-written:

*"As the $C_{20}$ Stokes coefficient is not well determined from GRACE mission, it is replaced with a more accurate solution from Satellite Laser Ranging (SLR) (Cheng and Ries, 2017)."*

**L98: Have you defined "GAD" earlier? Please provide details. Also, provide detailed justification of effects ignored here.**

The GAD acronym has not been defined. It stands for:
  G: Geopotential coefficients;
  A: Averaged of any background model over a time period;
  D: Bottom pressure over oceans, zero over land.
However, in the GRACE community it is not usually defined (we have had to check its meaning in the GRACE user's manual). On the other hand, it is very common and it is usually referred just as "GAD product". In any case, we give a description of what represents the GAD product for readers no familiarized with GRACE jargon.

**L97-100: This sentence is not very clear, rephrase for better clarity.**
**L99: What is MPIOM?**

This part has been re-written:

"On the other hand, the atmospheric, and some oceanic, effects on gravity change had beforehand been removed from the processing of the GRACE data by CSR, for de-aliasing purposes, according to the operational numerical weather prediction (NWP) model from ECMWF and to an unconstrained simulation with the global ocean general circulation model MPI-OM -Max-Planck-Institute Global Ocean/Sea-Ice Model- (Dobslaw et al. 2017). To recover the "true" ocean mass variability, we restore the removed signal on the oceans adding back the GAD product, which is set to zero on the continents."

**L105 to 115: I am not able to quite follow these sentences, perhaps because of my lack of expertise in GRACE. I'd suggest writing them in more details for readers who are not well- versed with GRACE estimates. In fact, I feel the entire paragraph can be re-written for more clearity.**

We have re-written these sentences:

*"GRACE's degree-0 Stokes coefficients $C_{00}$ is set identically to zero on the recognition that Earth's total mass (including the atmosphere) is constant. Then, any increase (decrease) of the water-mass budget of the atmosphere will be counteracted by a decrease (increase) of the same amount of water-mass in the surface. However, after the atmospheric and dynamic oceanic mass changes are corrected in GRACE data, the GRACE $C_{00}$ are still set to zero even though they should match the opposite of the removed signals. To restore the lost degree-0 signal, the GAD product (which is set to zero on the continents) must be added back to GRACE with averaged ocean signal set to zero, and then, the $C_{00}$ from an atmospheric model must be subtracted from GRACE data to force the Earth's total mass to be constant. Doing so, the GRACE data will account for the global exchange of water-mass between the Earth surface and atmosphere."*

**L126: Bootstrap replications of what variables? Are these timeseries data? L128-130: Please write these in more details. "subtracted" what from what? Please clarify.**

We have re-written this part:

*"The reported 95% confidence intervals and the correlation coefficients are evaluated using the stationary bootstrap scheme of Politis and Romano (1994) (with optimal block length selected according to Patton et al., 2009), and the percentile method. The intuition underlying the bootstrap is simple. Suppose that the observed time series $x_1, ..., x_n$ is a realization of the random vector $(X_1,..., X_n)$ with joint distribution $P_n$ and which is assumed to be part of a stationary stochastic process. Given $X_n$, we first build and estimate $\hat{P}_n$ of $P_n$. Then B random vectors $(X_1^*, ..., X_n^*)$ are generated from $\hat{P}_n$. If $\hat{P}_n$ is a good approximation of $P_n$, then the relation between $(X_1^*, ..., X_n^*)$ and $\hat{P}_n$ should well reproduce the relation between $(X_1,..., X_n)$ and $P_n$ (for an introduction of bootstrap methods for time series see Kreiss and Lahiri (2012) and the references therein). Here, the number of bootstrap replications was set to B=2000. In general, half length of the confidence interval can be very well approximated by twice the standard deviation of the sample mean estimated from the bootstrap replications. Prior to applying the bootstrap to a time series, least-squares estimated linear/quadratic trend and sinusoid with the most relevant frequencies are removed from it to meet the stationarity conditions of the method. In particular, each series has been decomposed into trend, seasonal and residual components. The bootstrap is applied to the residual component producing bootstrap samples of the residuals. For the evaluation of confidence intervals for the different components of WT, the trend and seasonal terms are added back (to the bootstrap sample of the residuals) producing bootstrapped time series of the component of interest. These samples are then used for further analysis. As an illustration, for the WT N component we proceed as follows: (i) a model with linear, annual, and semiannual signals is fitted to the data. The fitted linear trend and annual and semiannual signals are subtracted from the original time series; (ii) the stationary bootstrap is then applied to the residuals producing 2000 bootstrap samples of the residuals; (iii) The estimated trend and seasonal components are added back to each bootstrap sample of the residuals*

*obtaining an ensemble of 2000 bootstrapped time series for the N component; (iv) these 2000 bootstrapped time series are used to obtain 95% confidence intervals for the mean fluxes (average of N over the 14 year period of study) and for the amplitude and phase of the annual component using the percentile method. For the mean fluxes, the average of N for each of the 2000 bootstrapped time series was first evaluated and then the 0.025 and 0.975 percentiles of these 2000 averages were reported as 95% confidence interval. For the study of the climatology, a linear trend model with annual and semiannual components was fitted to the 2000 bootstrapped time series producing corresponding estimates of the annual amplitude and phase. The 0.025 and 0.975 percentiles of these estimates were reported as 95% confidence intervals.* In order to study the robustness of the results with respect to the model choice, the analysis is rerun using 11 alternative models obtained considering different forms for the trend component (quadratic or constant) and including higher frequencies in the harmonic regression (up to 5). The results are robust. The relative difference with respect to the reported values is smaller than 1.2% for point estimates and smaller than 3.3% for the extremes of the 95% confidence intervals.

*As an independent check of the bootstrap, confidence intervals for the mean value of N have been also evaluated by propagating the error estimate in GRACE data (using the JPL GRACE mascon solution for which error estimates are available). The resulting intervals were consistent with those of the bootstrap method. In particular (see Section 4 for details), we show that in all cases the bootstrap intervals contain the intervals obtained from error propagation. In this respect, the $CI_{95}$ from bootstrap analysis can be considered a conservative estimate. This should be expected, since the residual component underlying the bootstrap approach includes measurement errors and other type of errors (related, for example, with the estimate of the trend and seasonal terms). As a result, the uncertainties in the transports estimated by the bootstrap should be larger than the corresponding uncertainties estimated by error propagation.*

*Note that for the study of correlation the bootstrap was applied to the bivariate time series of the residuals of the two variables of interest producing an ensemble of 2000 bivariate time series of residuals. For each bivariate time series of residuals the correlation between the two components of the series was first evaluated. The average and the 0.025 and 0.975 percentiles of these 2000 estimates were reported as point estimate and confidence limits for the correlation between the two variables of interest (correlation between residual components is used to avoid spurious correlation)."*

**L135: All these can be written concisely in a table.**

The text in line 135 summarizes the robustness of the estimation of the main feature of the *N* component of WT, with respect to the model choice (trend + seasonality):

L135:  *In order to study the robustness of the results with respect to the model choice, the analysis is rerun using 11 alternative models obtained considering different forms for the*

*trend component (quadratic or constant) and including higher frequencies in the harmonic regression (up to 5). The results are robust. The relative difference with respect to the reported values is smaller than 1.2% for point estimates and smaller than 3.3% for the extremes of the 95% confidence intervals.*

Although we see the point of including a table we decided to maintain line L135 in its original form since, in our opinion, it provides a better summary of the robustness than a table. The table, in fact, would be quite large ( with height 11 times the height of the actual table 1) and should include details of each model (which might be different for the different basins). For each of the quantities of interest in table 1 the reader should compare the 11 solutions provided by the different models (each solution comprises a point estimate and a confidence interval). All these comparisons, in our opinion, are more effectively summarized in the two lines:

*"The relative difference with respect to the reported values is smaller than 1.2% for point estimates and smaller than 3.3% for the extremes of the 95% confidence intervals."*

**Figure 2: It might be useful to show N in right panels. Perhaps, you can only keep P- E, rather than showing both P and E.**

In the new version of Figure 2 N has been included in right panels.

**Figure 2: Should be made clearer. Shouldn't "R" be same in both left and right columns. Perhaps, you can remove the space between columns and extended the x-axis of the plots.**

Yes, *R* is the same in both columns. We have removed the R time series in the right column of Figures 2 and 4. With respect to the "design" of the figures, we have tried many options and selected the one that seems to us most clear (see the new Figure 2 below).

[Figure]

**Figure 2. WT of Equations (1) and (2) in the Pacific (first row), Atlantic/Indian/Arctic (AIA) oceans collectively (second row), and their drainage basins.** First column: associated land drainage basins; second column: ocean basins. Labels in the vertical axis correspond to the mean ± standard deviation of the associated curve. Thick lines are the low pass filtered signal by a Hann function of 24 months. All curves in the same panel are plotted on the same scale. *P, E,* and *P–E* are from ERA5 dataset; *dW* is estimated from GRACE; *R* and *N* are estimated as a residual in equations 1 and 2, respectively.

**L145: The P-E estimate of 142 Gt/month should be corrected for final presentation, since it was earlier noted that ERA5 may have some internal inconsistency when comparing previously estimated values from the literature.**

The global average of $P-E$ is 188 Gt/month and it should be $[-4.3, 0.9]$. The 142 Gt/month is for the Pacific Ocean and it may be corrected according to its area. However, we have not applied this correction for two reasons: (1) It is irrelevant for the estimation of $N$ after the degree-0 correction of GRACE data is applied; (2) We do not know how it would affect the data locally and how errors would spread on computations.

**L145: How does the results on fluxes in this manuscript compare with the values in the literature.**

See our answer to specific comments 2 and 3.

**Figure 3: I'm not sure if Figure 3 is required. I think it can be removed without loss of any critical details.**

We are very interested in showing the agreement between the Pacific outflow and the AIA inflow, which is not trivial at all. That is the reason to keep them in a separate figure.

In our opinion, the comparison between the Pacific outflow and the AIA inflow is critical as a warranty of consistency in the processing of the data.

**L145 to 155: I would suggest the values to be put in a graphical format, which might look more appealing that writing the numbers.**

It can be seen graphically now in the new diagram inserted as Figure 5 (see the answer to the specific comment 4).

**L164: What is basis of deciding the salinity of water in this study?**

We provide the results in the form of Gt/month. Freshwater from ERA5 is 1000 kg/m$^3$, and GRACE data are directly kg/m$^2$. When converting Gt/month into Sv, it does not matter if we choose a density 1020 or 1035 kg/m$^3$ since we only show two significant digits (in Sv).

**L181: I though the data is monthly; if so, how can we say day of month also?**

Yes, data are monthly. The day of the maximum annual signal is obtained from the annual component fitted to the data, which is an analytical function.

**L198: Again, it seems important that authors add N time series also in Figures 2 and 4.**

*N* component was already shown in Figure 4 and has been added in Figure 2.

**Figure 6: In addition to showing the timeseries, you can show the correlation values of each index with Pacific outflow.**

The correlation coefficients are shown in the new version of the Figure (see the answer to the specific comment 5).

**L225: The importance of these changes has not been discussed before in the manuscript.**

A previous comment has been added in the description of GRACE data:

> *"Any other non-surficial effect such as long-term tectonics would be incorrectly interpreted as water mass fluxes (Chao, 2016) but they may only have importance in the determination of secular trends; so are the non-climatic sources such as the rare, local earthquake events."*

**Appendix:**
**Have we referend to Appendix anywhere in the main text?**

It was mis-referenced at the end of the third paragraph of the section "Discussion and Conclusions" as Supplementary Material. It has been corrected.

> *"However, the mean freshwater flux in the Pacific (1261 Gt/month) quite mis-matches that in the Atlantic/Indian Oceans (−1837 Gt/month), meaning that the approximation was quite poor and hence the N term was not properly estimated in these studies (see* Appendix *for further discussion)."*

---

## Author Comment (AC3) · 3 Sep 2020

**Answer to Reviewer 3**

We thank the reviewer for thorough reading and thoughtful comments and suggestions. A detailed discussion of the changes that we made in response to the reviewer's comments is given below. In what follows, we state the reviewer's comment in boldface, and describe our response in plain text. Text in the manuscript is represented in italics. The text that has been modified/included in the new version has been highlighted in red.

**Review for esd-2020-54 "Water transport among the world ocean basins within the water".**
**In general, this study is interesting to me, with the global ocean basins to study water mass transport based on GRACE and ERA5. The conclusions are generally supported by the data analyses in this study, but more validations/evaluations are necessary to improve the reliability of the results. At least, a careful intercomparsion between this study and previous studies/literatures can be discussed to enhance our understanding.**

We have included several improvements in the manuscript:

1. **Comparison with similar studies:** As far as we are applying a new methodology, there are not many studies to compare with. The only two studies, up to our knowledge, doing something similar are discussed in the third paragraph of the section "Discussion and Conclusions":

   *"The results presented here are consistent with the well-known salinity asymmetry between the Pacific and Atlantic Oceans (Reid, 1953; Warren, 1983; Broecker et al., 1985; Zaucker et al., 1994; Rahmstorf, 1996; Emile-Geay et al., 2003; Lagerloef et al., 2008; Czaja, 2009; Reul, 2014). However, they are in contrast to previous GRACE-based studies where a simple seesaw WT between the Pacific and the Atlantic/Indian oceans was reported [Chambers and Willis, 2009; Wouters et al., 2014]. In those studies, the P−E+R term in Equation 2 in both Pacific and Atlantic/Indian Oceans was approximated by that from the global ocean mean. However, the mean freshwater flux in the Pacific (1261 Gt/month) quite mis-matches that in the Atlantic/Indian Oceans (−1837 Gt/month), meaning that the approximation was quite poor and hence the N term was not properly estimated in these studies (see Appendix for further discussion)."*

   As stated in the text, in the Appendix we explain in details why the proposed methodology overcomes some important limitations of previous approaches which will always show a seesaw of water transport, even if it does not exist.

2. **Other datasets:**

   The *P*, *E*, *P−E*, and *R* components are auxiliar in this study. However, we have added some more references for comparison purposes in the last paragraph of Section 3.1:

*"Corresponding analyses have been performed for the Atlantic, Indian, and Arctic Oceans separately. The time evolution of the WT components in Eqs. 1 and 2 are shown in Figure 4, and a diagram of the water-mass fluxes is shown in Figure 5. On average, the Atlantic Ocean receives 926 Gt/month ($CI_{95}$=[876, 980]; or 0.36 Sv) of salty water, and loses to the atmosphere 879 Gt/month ($CI_{95}$=[828, 930]) via P−E+R. The latter is equivalent to a freshwater deficit of 0.34 Sv, which increases the near-surface salt concentration and enables water to sink in North Atlantic producing deep water. These values are close to the 0.13-0.32 Sv estimated from ocean models, as needed to keep salinity balance in the Atlantic Ocean (Zaucker et al., 1994). Similarly, the Indian Ocean loses 957 Gt/month ($CI_{95}$=[894, 1022]) of freshwater that is restored by 991 Gt/month ($CI_{95}$=[907, 1073]) of salty water. The freshwater lost via P−E+R by the Atlantic and Indian Oceans goes to the Pacific (1261 Gt/month, $CI_{95}$=[1171, 1347]) and Arctic (730 Gt/month, $CI_{95}$=[712, 747]) Oceans, which return 1194 ($CI_{95}$=[1096, 1291]) and 723 ($CI_{95}$=[708, 739]) Gt/month of salty water through the ocean, respectively. Then, the Pacific presents a surplus of freshwater that reduces near-surface salt concentration, which prevents the formation of deep water. Together, the Pacific and Arctic Oceans supply 1917 Gt/month ($CI_{95}$=[1812, 2021]) of water to the Atlantic and Indian Oceans, where it is reincorporated into the water cycle via net E−P. As in previous studies (see Craig et al., 2017 for a synthesis), the freshwater lost in the Indian Ocean is similar to that in the Atlantic Ocean. In those studies, P-E+R is close to zero in the Pacific Ocean, producing a difference of 0.4 Sv between Atlantic and Pacific Oceans. In this study, P-E+R is 1261 Gt/month in the Pacific Ocean and the difference with the Atlantic increases to ~0.8 Sv. Some of these differences would be expected as far as the ocean basins are not defined in exactly the same way. On the other hand, the global R is 3781 Gt/month (or 3781 × 12 = 45368 $km^3$/year), close to the 41867 $km^3$/year reported by the Global Runoff Data Centre (GRDC, 2014). At basin scale, R is 16834 $km^3$/year in the Pacific, greater than the 11826 $km^3$/year reported by GRDC. In the Atlantic, Indian, and Arctic, R is 18228, 4479, and 5827 $km^3$/year, respectively, which is closer to the GRDC values: 20772, 5238, and 4080 $km^3$/year. Finally, according to the diagram in Figure 5, the water content in the atmosphere decreases 178 Gt/month (and it is gained by Earth's surface), but this amount is not realistic as discussed in Section 2 since it should increase a few Gt/month [Nilsson and Elgered, 2008]. This value differs from the 188 Gt/month mentioned in Section 2 because the endorheic regions are not accounted here."*

More importantly, we have extended our analysis to other datasets. The objective is to show that our main results concerning the *N component*, are not an artifact of CSR GRACE and ERA5 datasets. As a result, there is new section entitled "Comparison with other datasets":

*"Equations 1 and 2 are applied to estimate the Pacific outflow using different datasets:*

*(1) CSR GRACE mascon solution is replaced by the JPL GRACE mascon solution provided by the Jet Propulsion Laboratory/NASA (Watkins et al., 2015; Wiese et al., 2019). Similarly to CSR data, JPL are corrected for GIA effects, $C_{20}$ Stoke coefficients*

*are replaced by a solution from SLR, and data are reduced to 1º regular grids from 0.5º regular grids. Besides, we have applied the degree-0 Stoke coefficients correction. However, CSR and JPL mascon solutions are not directly comparable. The main reason is that an estimate of degree-1 coefficients has been added to JPL mascon solutions, and the GAD product has not been added back. The corrections applied by JPL are not supplied separately and we cannot do/undo any of the corrections to process JPL data as we did with CSR data. In particular, the GAD product is not available for JPL. In any case, the JPL solution is useful here since it provides an error estimate of the mascon solution that can be propagated to obtain confidence intervals of N, which are independent from those estimated with the bootstrap analysis. Table 2 shows the $CI_{95}$ of the mean values of the N component for different ocean basin estimated from error propagation and bootstrap analysis. It is observed that in all cases the $CI_{95}$ from error propagation are included in those from bootstrap analysis, meaning that the latter are a conservative estimate of the error. JPL propagated error can be expected to be similar to that propagated from CSR error estimates (which are not available), and then we can assume that the reported $CI_{95}$ for N calculated from CSR data are a conservative estimate. Besides, comparing Tables 1 and 2, it is observed that the mean values of N are quite similar and that the $CI_{95}$ largely overlap. Regarding to the time variability, the values of the N component from CSR and JPL mascon solutions show Pearson correlation coefficients greater than 0.85 (p-value < $10^{-3}$), except for the Atlantic (0.70). Thus, despite the different processing of CSR and JPL data, the reported analysis for the N component is robust with respect to the choice of GRACE datasets.*

**Table 2. Mean net WT from JPL mascon for different ocean basins according to Equation 2 .** $CI_{95}$ are estimated as propagation of mascon errors provided by JPL, and from bootstrap analysis. Units are Gt/month.

| | | Mean ($CI_{95}$ from error propagation) | Mean ($CI_{95}$ from bootstrap) |
|---|---|---|---|
| Outflows | Pacific | 1182 (1143, 1220) | 1182 (1062, 1306) |
| | Arctic | 735 (713, 757) | 735 (711,761) |
| | Pacific + Arctic | 1917 (1872, 1961) | 1917 (1806, 2036) |
| Inflows | AIA | 1183 (1092, 1274) | 1183 (1077, 1282) |
| | Atlantic | 919 (866, 972) | 919 (845, 985) |
| | Indian | 999 (980, 1018) | 999 (928, 1067) |
| | Atlantic + Indian | 1918 (1862, 1974) | 1918 (1838, 2003) |

*(2) ERA5 P and E data are replaced by several datasets for comparison purposes. The objective is not to be exhaustive in the selection, but rather to show that the reported features of the N component are quite robust with respect to the choice of the P and E datasets. The data sets considered are:*

*(i) Continental P from GPCC (Schneider et al., 2011), GPCP (Adler et al., 2018), CMAP (Xie and Arkin, 1997), UDel (Willmott and Matsuura, 2001), and GLDAS/Noah (Rodell et al., 2004; Beaudoing and Rodell, 2016).*
*(ii) Ocean P from GPCP and CMAP.*
*(iii) Continental E from GLEAM (Miralles et al., 2011; Martens et al., 2017) and GLDAS/Noah.*

*(iv) Ocean E from OAFlux (Yu et al., 2008) and HOAPS/CM SAF (Schulz et al., 2009).*

*The Pacific outflow is estimated with the 162 possible combinations of P and E, including ERA5. The time period is 2003-2016, except for HOAPS/CM SAF and GPCP, which span from 2003 to 12/2014 and 10/2015, respectively. The degree-0 corrections in GRACE data is made for each combination. Note that only ERA5 includes P and E for both continents and oceans. All grids have been homogenized to 1° regular grids. The main concern here is the heterogeneity of the spatial coverage among datasets. To make the results comparable among datasets, the computations are restricted to the common grid points, which do not cover the entire Earth (Figure 8a).*

[Figure]

**Figure 8. Monthly time series of (the opposite of) the Pacific outflow estimated from 162 combinations of *P* and *E* datasets.** a) Spatial coverage common to all datasets. b) Pacific outflows: Gray thin curves are the 162 Pacific outflows estimated in the common grid points to all datasets (no global coverage); black and red curves are based on ERA5 *P* and *E* and are obtained using either only the grid points common to all datasets (black curve) or global coverage (red curve). Note that the red curve is the same as in Figure 3.

*However, in spite of the fact that due to the partial coverage the principle of water mass conservation is not accomplished, the Pacific outflow obtained in the common grid points from ERA5 (black line in Figure 8b) is quite in agreement with the same signal obtained with global coverage (red line in Figure 3 which is also reported as red line in Figure 8b). The Pearson correlation coefficient between the two signals is 0.994 (p-values < $10^{-3}$) with an average difference around 50 Gt/month. In general, the Pacific outflows estimated from all the P and E dataset combinations show qualitatively the same signal than the one reported in Figure 3. For each of the 162 estimates of the Pacific outflows corresponding to the possible P and E dataset combinations, we evaluated the average outflow (over the period of study), which is 968 Gt/month (STD: 489), and the correlation with the Pacific outflows in Figure 3, which is 0.82 (STD: 0.06; p-values < $10^{-3}$).*

*These experiments show that the reported net WT are physically consistent among datasets, at least qualitatively."*

3. **Bootstrap Vs Error propagation:** The confidence intervals estimated from bootstrap have been compared to those estimated from error propagation of the GRACE mascon. As CSR mascon solution does not provide such error estimates, we have used the JPL mascon solution for the comparison. An explanation of why bootstrap confidence intervals contains, as expected, the error propagation confidence interval has been also provided. In the description of the bootstrap method we have included the following text:

*As an independent check of the bootstrap, confidence intervals for the mean value of N have been also evaluated by propagating the error estimate in GRACE data (using the JPL GRACE mascon solution for which error estimates are available). The resulting intervals were consistent with those of the bootstrap method. In particular (see Section 4 for details), we show that in all cases the bootstrap intervals contain the intervals obtained from error propagation. In this respect, the $CI_{95}$ from bootstrap analysis can be considered a conservative estimate. This should be expected, since the residual component underlying the bootstrap approach includes measurement errors and other type of errors (related, for example, with the estimate of the trend and seasonal terms). As a result, the uncertainties in the transports estimated by the bootstrap should be larger than the corresponding uncertainties estimated by error propagation.*

The details are shown in the new section "Comparison with other datasets", that can be found above.

**4. About the lack of correlation.** We have included a discussion on the lack of correlation between the inter-annual transports and the indices of ocean-atmosphere interaction. In particular we propose the two following explanations:

*"To explore this lack of correlation, we have estimated the correlation coefficient between each climatic index and each WT component (Figure 7b).*

[Figure]

[Figure]

**Figure 7. Pacific outflow and climatic indices for ENSO, AMO, AO, and AAO. a)** Time series of Pacific outflow is de-trend and de-season. All time series are normalized to have unit variance. Values in the parenthesis are the correlation coefficient between the corresponding climatic index and the Pacific outflow. b) Correlation coefficients between de-trend and de-season WT components of different regions and the climatic indices.

*All of them are lower than 0.3 except for 6 cases in 2 regions. In the Arctic, P and P−E in the drainage basins of the Arctic show a correlation of ~0.5 with the AO. This correlation is natural since that is the area of influence of the AO. The other region is the Pacific, where, as expected, the SOI shows a correlation around 0.5 with P, P−E, and R in the drainage basins, and around -0.4 with P in the ocean. However, this individual correlation does not extend to the Pacific outflow. In order to understand why this is the case, it is convenient to express the N component of the water transport as a function of (P-E) and dW. According to Equations 1 and 2 we have:*

$$N = -(P-E)_{ocean} - R + dW_{ocean} =$$
$$\underbrace{-(P-E)_{ocean}}_{X_1} \underbrace{-(P-E)_{land}}_{X_2} + \underbrace{dW_{land}}_{X_3} + \underbrace{dW_{ocean}}_{X_4}.$$
(3)

*It can be shown that the correlation between N and a given index can be express as follows*

$$corr(N, Index) = \sum_{i=1}^{4} corr(X_i, Index) \cdot \frac{std(X_i)}{std(N)},$$
(4)

*where corr denotes the correlation coefficient, and std stands for standard deviation. As shown in equation (4), the correlation between N and a given index is a linear combination of the correlation between each component and the index. The coefficients of the linear combination $std(X_i)/std(N)$ are proportional to the standard deviation of each component. The components of equation (4) for the Pacific outflow and the SOI index are shown in Table 3. Despite the fact that some of the individual component exhibits significant correlation with SOI (in particular P-E in land and ocean) when combined with the corresponding coefficients their effects are canceled out yielding to a negligible correlation between water transport and SOI (below 0.03 in magnitude).*

*Another possible reason for the lack of correlation resides in the definition of the studied regions, for which the presence of subregions with positive and negative influence of an index results in an overall negligible/attenuated influence of the index in the overall region. For example, a positive phase of the AMO is related to an increase of P in western Europe (Sutton and Hodson, 2005), and the Sahel (Folland et al., 1986; Knight et al., 2006; Zhang and Delworth, 2006; Ting et al., 2009), but to a decrease of P in the U.S. (Enfield et al., 2001; Sutton and Hodson, 2005), and northeast Brazil (Knight et al., 2006; Zhang and Delworth, 2006). All these regions are included in the Atlantic drainage basin, and then the influence of a positive phase of the AMO is attenuated."*

**Table 3. Correlation coefficients between SOI and de-trend and de-season WT components involved to estimate the Pacific outflow according to Equations 3 and 4.**

| | $std(X_i)$ (Stand. Deviation) | $corr(X_i, SOI)$ (Correlation between $X_i$ with SOI) | $\dfrac{std(X_i)}{std(N)}$ (Coefficients) | $corr(X_i, SOI) \cdot \dfrac{std(X_i)}{std(N)}$ (Correlation ·Coefficient) |
|---|---|---|---|---|
| $X_1 = -(P{-}E)_{ocean}$ | 605 | 0.25 | 0.57 | 0.14 |
| $X_2 = -(P{-}E)_{land}$ | 212 | -0.53 | 0.20 | -0.11 |
| $X_3 = dW_{land}$ | 96 | 0.048 | 0.09 | 0.004 |
| $X_4 = dW_{ocean}$ | 711 | -0.10 | 0.67 | -0.07 |
| | Corr($N$,SOI) | | | -0.03 |

Note that table 3 provides also some insights about the causes of the interannual variability of Pacific Ocean outflow. The largest standard deviation of $P\text{-}E$ and $dW$ in the ocean suggests that these two components might drive the interannual variability of the Pacific Ocean outflow. This is confirmed by a correlation analysis. The correlation between $N$ and the $(P\text{-}E)_{ocean}$ is -0.70. The correlation between $N$ and the $dW_{ocean}$ is 0.84. The correlation of $N$ with the corresponding land components is below 0.18. In all cases, prior to the evaluation of the correlation the corresponding time series have been de-trend and de-season.

5. **Improvement in the visualization of main results**. We have included a new Figure with a diagram of the mean WT components to ease the reading:

[Figure]

**Figure 5. Diagram of the mean values of the WT of the studied regions.** Units are Gt/month.

**Other comments:**
**L44, change 'de' to 'the'.**

Typo has been corrected.

**In section 2: Methodology and Data: please use subtitles to re-organize the section, and improve the readability.**

Section 2 is now divided in 4 subsections:
*2.1 Methodology*
*2.2 Precipitation and Evaporation data*
*2.3 Time-variable GRACE data*
*2.4 Confidence intervals*

**Fig 1: suggest to add legends to indicate the locations of different ocean basins.**

Figure 1 has been modified, now it includes the names of the basins in the figure itself.

[Figure]

**Figure 1.** Pacific, Atlantic, Indian, and Arctic Ocean basins and their associated continental drainage basins according to the global continental runoff pathways scheme of Oki and Sud (1998). Within each basin, darker colour represents the continental basin, lighter colour the ocean basin. White regions represent endorheic basins.

**Fig 2: are these values obtained from GRACE? or a combination of ERA5 and GRACE? please briefly clarify this in the figure caption.**

The next sentence has been added to the caption:
*"P, E, and P–E are from ERA5 dataset; dW is estimated from GRACE; R and N are estimated as a residual in equations 1 and 2, respectively."*

**Fig 3: black curve is the AIA inflow: 'if' should be 'is'.**

Typo has been corrected.

**Fig 4: N component should be briefly explained in the caption.**

The last sentence of the caption is now:
*"Black lines are the WT N component, which are estimated as residuals in Equation 2."*

**Fig 6: please indicate the correlation coefficient R between each climate index and the Pacific outflow, at each time series.**

It has been included. See the new Figure 7b in the point 4 (About the lack of correlation) of this response.